# BMAL1 modulates glutamine supply to control haematopoietic stem and progenitor cell expansion

Tim Petzold[1,2,3], Lydia K. Lutes[1], Keila Navarro I. Batista[1], Antonia Konle[2], Bastien Baechler[1], Stéphane Jemelin[1], Holger Gerhardt[2,3,4], Rachel Golub[5], Christoph Scheiermann[1,6,7,*] and Julien Y. Bertrand[1,7,*,‡]

## ABSTRACT

Following specification in the dorsal aorta, haematopoietic stem and progenitor cells (HSPCs) proliferate in the HSPC niche, known as the caudal haematopoietic tissue (CHT) in zebrafish. Here, we demonstrate that *bmal1a*, a core component of the circadian clock machinery, is expressed in CHT endothelial cells (ECs) and affects HSPCs in a non-cell autonomous manner. Using endothelial cell-specific dominant-negative Bmal1a zebrafish lines, we demonstrate a striking increase in HSPC numbers in the CHT, resulting from enhanced HSPC proliferation. RNA-sequencing of dominant-negative *bmal1a* ECs sorted from the CHT shows a downregulation of *glud1a*, resulting in increased glutamine levels in the CHT. This newly discovered *bmal1a-glud1a*-glutamine pathway fuels HSPC expansion. We demonstrate that this glutamine synthesis pathway controlling HSPC expansion is likely conserved in the mouse fetal liver (FL) niche, in which hepatocytes are the likely source of glutamine. Together, our data uncover a previously unreported mechanism of HSPC homeostasis, in which EC BMAL1, expressed by the niche, controls the amount of bioavailable glutamine for HSPCs by regulating the expression of genes involved in glutamine synthesis.

**KEY WORDS: HSPC, Niche, Expansion, Glutamine, Endothelial cell, Zebrafish, Mouse**

[1]University of Geneva, Faculty of Medicine, Department of Pathology and Immunology, Rue Michel-Servet 1, Geneva 4, Switzerland. [2]Integrative Vascular Biology Laboratory, Max Delbrück Center (MDC) for Molecular Medicine in the Helmholtz Association, Robert-Rössle-Strasse 10, 13125 Berlin, Germany. [3]DZHK (German Center for Cardiovascular Research), 10785 Berlin, Germany. [4]Charité - Universitätsmedizin Berlin, 10117 Berlin, Germany. [5]Institut Pasteur, Université Paris Cité, INSERM U1223, Lymphocyte and Immunity Unit, 75015 Paris, France. [6]Biomedical Center (BMC), Institute for Cardiovascular Physiology and Pathophysiology, Walter Brendel Center for Experimental Medicine (WBex), Faculty of Medicine, Ludwig-Maximilians-Universität (LMU) Munich, Planegg-Martinsried, 82152 Munich, Germany. [7]Geneva Centre of Inflammation Research, University of Geneva, Faculty of Medicine, Rue Michel-Servet 1, Geneva 4, Switzerland.
*These authors contributed equally to this work

‡Author for correspondence ( julien.bertrand@unige.ch)

T.P., 0000-0001-9788-6167; L.K.L., 0000-0001-6133-6294; K.N.I.B., 0000-0001-9462-1817; H.G., 0000-0002-3030-0384; R.G., 0000-0002-2306-5363; C.S., 0000-0002-9212-0995; J.Y.B., 0000-0001-6570-4082

## INTRODUCTION

Circadian rhythms are biological oscillations that regulate most physiological and behavioural processes, allowing organisms to adapt to a fluctuating environment. In mammals, circadian rhythms are sustained by transcriptional translational feedback loops (TTFL) (Hurley et al., 2016). The core mammalian clock genes include *Bmal1* and *Clock*, which encode activator proteins of the TTFL. BMAL1 and CLOCK proteins form a heterodimer complex that binds to E-Box sequences in the promoters of target genes, including the TTFL repressor circadian genes *Per* and *Cry*. Transcription of Per and Cry genes results in heterodimerisation of the PER/CRY transcription factor complex, which ultimately inhibits *Bmal1* and *Clock* expression (Scheiermann et al., 2013). Together with additional auxiliary *Rev-erb*a (*Nr1d1*) and Ror regulatory loops of *Bmal1* and *Clock*, this TTFL mediates approximately 24 h oscillations (Relógio et al., 2011; Ueda et al., 2005).

The zebrafish circadian clock architecture is very similar to that of mammals. However, due to a third round of teleost whole-genome duplication in this species, additional copies of many genes are present, including those of the circadian clock (Liu et al., 2015). Zebrafish therefore possess two paralogues of *bmal1* (Wang, 2009), three of *clock* (Wang, 2008b), four of Per genes (Wang, 2008a) and seven of Cry genes (Liu et al., 2015), adding molecular complexity to the zebrafish clock. The precise functions of the individual gene paralogues have yet to be established (Froland Steindal and Whitmore, 2019).

BMAL1 and CLOCK proteins regulate the transcription of thousands of target genes. In mammals, it is thought that BMAL1 and CLOCK drive the expression of 15% of the transcriptome (Trott and Menet, 2018), including a plethora of genes involved in metabolism (Reinke and Asher, 2019). In fact, metabolism, including cellular and organellar metabolism (Neufeld-Cohen et al., 2016), as well as blood nutrient homeostasis (Lamia et al., 2008), is one of the most rhythmic processes. Light-dark cycles have been shown to regulate haematopoietic stem and progenitor cell (HSPC) self-renewal and differentiation by cell-intrinsic metabolic re-wiring of these cells (Golan et al., 2018). Furthermore, it has previously been demonstrated that many genes involved in metabolism are under the control of BMAL1 and CLOCK in malignancies such as acute myeloid leukaemia (AML) (Puram et al., 2016). The expression of genes involved in the synthesis of amino acids such as glutamine have been shown to be altered in the absence of a functional circadian clock in the context of cancer (Wang et al., 2024), while emerging evidence also shows that circadian clock genes are expressed in oocytes and during the development of the rat embryo and primate fetus (Seron-Ferre et al., 2007), suggesting that these genes play a role during embryogenesis.

In vertebrates, haematopoiesis takes place in successive waves, starting with primitive haematopoietic cells, followed by the

**DEVELOPMENT**

emergence of erythro-myeloid progenitors, and culminating with the definitive wave in which haematopoietic stem and progenitor cells (HSPCs) are generated (Bertrand and Traver, 2009). HSPCs are specified from the dorsal aorta (Bertrand et al., 2010a; Boisset et al., 2010; Kissa and Herbomel, 2010), and many recent studies have demonstrated heterogeneity in terms of developmental potential among these aorta-derived progenitors. Work in zebrafish has, for example, pointed to the existence of lymphoid-restricted progenitors (Tian et al., 2017), lymphoid-erythroid primed progenitors and lympho-myeloid primed progenitors, among others (Ulloa et al., 2021; Torcq et al., 2025). The identities of such progenitors were determined using transcriptomics analysis of sorted progenitors, or *a posteriori*, after lineage tracing; currently, there is a lack of transgenic marker lines that would allow these sub-populations to be robustly characterised *in vivo* in zebrafish. Among these HSPCs, only a rare subset fulfils the criteria of true bona fide haematopoietic stem cells (HSCs) that can contribute to adult multilineage haematopoiesis (Bertrand et al., 2010a; Tian et al., 2017; Henninger et al., 2017). However, all these definitive progenitors share a number of common features: they are derived from the aorta, their generation is dependent on Notch signalling and they all depend on the expression of *gata2b* (Butko et al., 2015). Following specification from the aorta, HSPCs colonise the caudal haematopoietic tissue (CHT) in zebrafish, or the fetal liver (FL) in mammals, via the blood circulation (Mahony and Bertrand, 2019). A crucial component of this embryonic niche is the vascular endothelium (Tamplin et al., 2015; Khan et al., 2016), which provides HSPCs with highly regulated signals allowing them to expand. While some factors controlling the vascular HSPC niche have been discovered (Cacialli et al., 2021; Mahony et al., 2016; Xue et al., 2017), a complete picture of the molecular interplay involved is lacking. In particular, the role of the circadian clock in this process is unknown.

Previous work has shown that circadian clock genes are expressed in venous ECs of the zebrafish embryo at 24 h post-fertilisation (hpf) (Gurung et al., 2022). We therefore reasoned that the clock may play a role in HSPC development in the CHT. Here, using newly generated EC-specific dominant-negative zebrafish lines, we report that the core circadian clock component *bmal1a* is a previously unknown regulator of HSPC expansion in zebrafish. We demonstrate that endothelial *bmal1a* acts as a negative regulator of HSPC proliferation in the vascular niche during zebrafish development, by controlling the expression of *glud1a*, a gene that plays a key role in the glutamine synthesis pathway. Glutamine plays an important role in cell proliferation (Yoo et al., 2020). In the context of haematopoiesis, glutamine has been shown to promote myeloid differentiation, augmenting the number of myeloid colonies in *in vitro* culture assays (Dass et al., 1984). More recently, glutamine has been shown to be important for emergency myelopoiesis in the context of systemic inflammation, by promoting myeloid cell expansion and differentiation (Pizzato et al., 2023). In this process, glutamine is converted into glutamate, then into α-ketoglutarate, which can fuel the Krebs cycle in myeloid progenitors (Pizzato et al., 2023). Finally, glutamine is also important for erythroid development, since its conversion into succinyl-CoA is essential for haem production (Burch et al., 2018). This haem production, however, produces toxic ammonium, which causes oxidative stress (Lyu et al., 2024). Detoxification of ammonium is brought about through the conversion of glutamate into glutamine via glutamine synthetase, which is upregulated in erythroid precursors (Lyu et al., 2024). Therefore, glutamine acts as an important amino acid in both steady-state and malignant haematopoiesis. As such, targeting glutamine-related pathways may provide novel treatment routes for AML and related syndromes (Xiao et al., 2023).

Our data indicate that *bmal1a* controls the expansion rate of HSPCs, by regulating the glutamine concentration in the niche. While deletion of *Bmal1* in mouse ECs did not result in a similar HSPC phenotype, this is likely due to *Glud1* being exclusively expressed by hepatocytes in the mammalian FL. When we increased the activity of Glud1 in *in toto* FL organ culture through addition of L-leucine, we observed a significant decrease in HSPC numbers, demonstrating that the role of this important metabolic pathway for HSPC expansion is also conserved in the mouse haematopoietic niche. Taken together, we have discovered a previously unknown, non-cell autonomous molecular mechanism controlling HSPC homeostasis: Bmal1 reprograms glutamine metabolism by regulating genes involved in glutamine synthesis, which governs the rate of HSPC expansion.

## RESULTS

### The core circadian clock gene *bmal1a* is expressed in the CHT vasculature

To investigate the spatio-temporal expression patterns of circadian clock genes during zebrafish development, embryos were raised in 12 h light:dark cycles before whole-mount *in situ* hybridisation (WISH) for *bmal1a*, *bmal1b* and *clocka* was performed every 6 h between 24 and 84 hpf (*in situ* images shown between 24-66 hpf). *bmal1a* was expressed in the CHT region between 24 and 42 hpf, but not at later stages (Fig. 1A,B). Furthermore, *bmal1a* expression in this region did not oscillate in a circadian manner (Fig. 1B). However, *bmal1a* showed oscillatory expression in the heads of embryos in the assessed timeframe, indicating tissue-specific expression patterns (Fig. S1). Compared to *bmal1a*, *bmal1b* was more broadly expressed along the entire trunk at 24 hpf (Fig. S2A), whereas *clocka* expression was specifically expressed in the CHT at 24 hpf (Fig. S2B). This indicates that the protein products of *bmal1a* and *clocka* may functionally interact in that region.

Since mouse *Bmal1* is the only non-redundant circadian clock gene (Scheiermann et al., 2018), we focused our subsequent investigations on *bmal1a* and *bmal1b*. In order to determine whether ECs are responsible for the expression observed by *in situ* hybridisation, we sorted ECs from *kdrl:EGFP* embryo tails at 24 hpf by fluorescence-activated cell sorting (FACS; for gating strategy, see Fig. S2C), before carrying out *bmal1a* and *bmal1b* qPCR (Fig. 1C, Fig. S2D). Expression of *bmal1a* and *bmal1b* were both found to be significantly enriched in tail ECs with *bmal1a* being more highly expressed than *bmal1b* (Fig. 1C, Fig. S2D). However, to avoid any potential functional redundancy, we designed a strategy that would impair the activities of both Bmal1a and Bmal1b proteins in CHT-ECs, to study their contribution(s) to the haematopoietic niche.

### Endothelial-specific dominant-negative Bmal1a results in a non-cell autonomous increase in HSPCs in the CHT

To specifically impair Bmal1a activity in ECs, we engineered a new dominant-negative (DN) *bmal1a* zebrafish line (*UAS:DN-bmal1a*). This DN *bmal1a* encodes a protein that contains the DNA-binding domain, a PAS domain required for heterodimerisation to Clock and a nuclear translocation domain (Fig. S3A). However, it lacks the C-terminal transactivation domain required by Bmal1a to induce transcription of its target genes (Gustafson et al., 2017). We reasoned that the *DN-Bmal1a* protein generated in the *UAS:DN-bmal1a* line would also occupy E-box elements and, as such, would prevent endogenous Bmal1 and Clock proteins from binding these DNA motifs.

We crossed the *UAS:DN-bmal1a* with *kdrl:GAL4* zebrafish adults, to specifically express *DN-bmal1a* in embryonic ECs. In

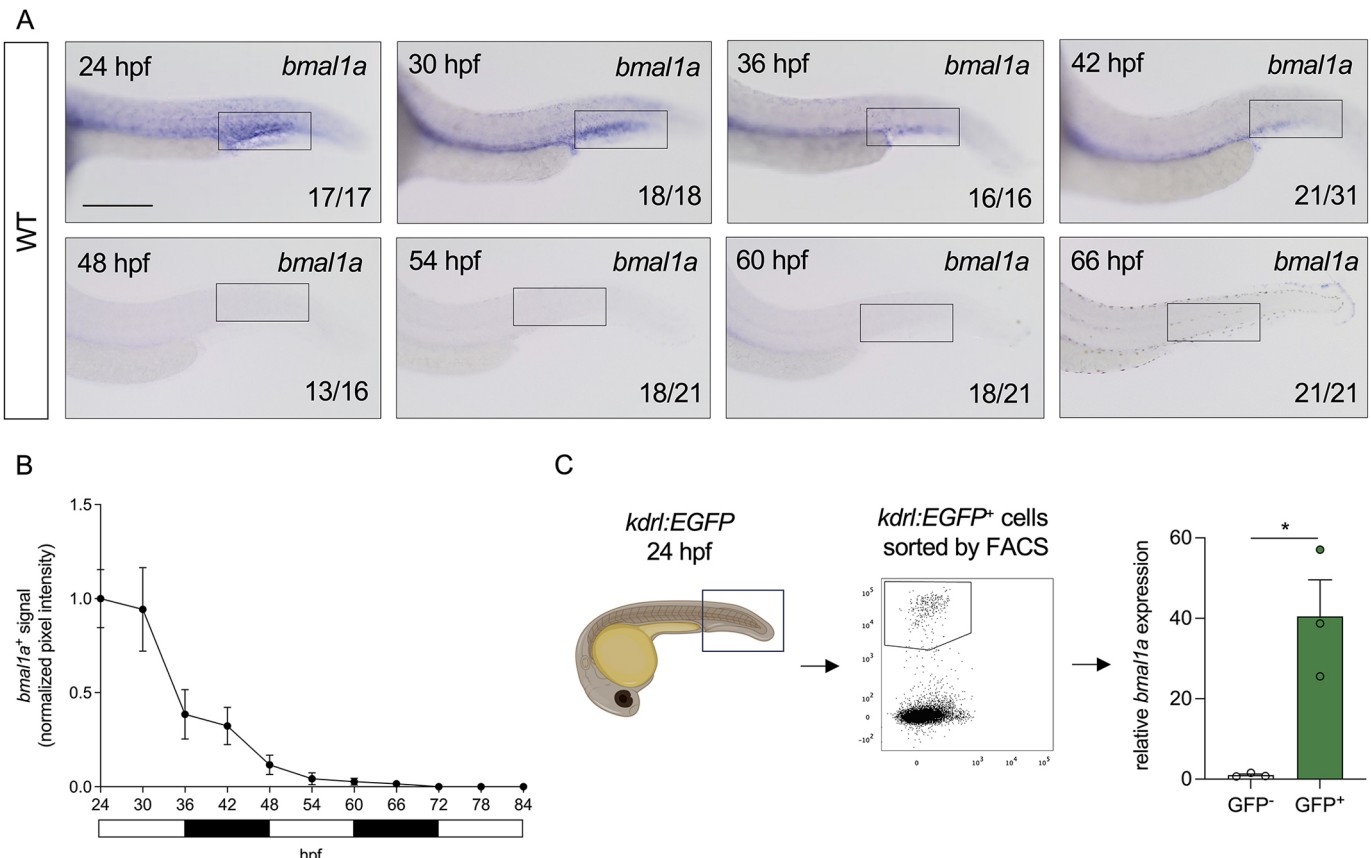

**Fig. 1. _bmal1a_ is expressed in endothelial cells of the caudal haematopoietic tissue but its expression in this region is not rhythmic in embryos raised in light-dark cycles.** (A) _In situ_ hybridisation of _bmal1a_ from 24-66 hpf in the tails of wild-type zebrafish embryos raised in light-dark cycles. (B) Quantification of normalised _bmal1a in situ_ hybridisation signal between 24 and 84 hpf in the caudal haematopoietic tissue (CHT) of wild-type zebrafish raised in light-dark cycles. _bmal1a_ expression diminishes over time and is not rhythmic; $n$=9 or 10 embryos for each timepoint. (C) _kdrl:EGFP_ tails were dissected at 24 hpf, before endothelial cells were isolated by FACS and _bmal1a_ expression was quantified by qPCR. _bmal1a_ expression is enriched in endothelial cells; $n$≈50 embryos in triplicate. Statistical significance of the differences between the two groups was calculated using unpaired two-tailed Student's _t_-tests assuming equal variance (*$P$<0.05). Rectangles in A indicate the CHT area. Scale bar: 200 μm. White and black rectangles in B denote light and dark periods. Created in BioRender by Petzold, T. (2026) https://BioRender.com/y7gmtpg. This figure was sublicensed under CC-BY 4.0 terms.

_kdrl:GAL4;UAS:DN-bmal1a_ embryos, we detected _bmal1a_ throughout the whole vascular system, as determined by WISH (Fig. S3B), in contrast to CHT-specific _bmal1a_ expression in wild-type controls. No gross morphological alterations were present in these double transgenic embryos. In particular, aortic vascular specification and development were unaffected in _kdrl:GAL4;UAS:DN-bmal1a_ embryos, as indicated by normal _dll4_ expression at 28 hpf (Fig. S3D) and _kdrl:EGFP_ expression at 48 hpf (Fig. S3E). To determine whether _DN-bmal1a_ efficiently disrupted circadian clock function, we performed qPCR analysis of the expression of _per2_, a known Bmal1:Clock target gene and component of the circadian architecture in both mammals (Takahashi, 2017) and zebrafish (Ruggiero et al., 2021). _per2_ mRNA expression was decreased in _kdrl:GAL4;UAS:DN-bmal1a_ embryos compared to controls at 36 hpf (Fig. S3C), suggesting that the construct was indeed acting as a dominant negative. _per2_ expression was not totally abolished, as our dominant-negative construct was effective only in ECs, while qPCR was performed on whole embryos.

We also generated a second _DN-bmal1a_ line that targets the highly-conserved E-K-R-R motif, which is required for binding to the E-box consensus sequence CANNTG in the promoters of target genes (Hosoda et al., 2004). Substitution of the E-K-R-R arginine at position 91 (underlined) with alanine in the mouse, results in a dominant-negative BMAL1, since it is unable to support DNA

binding, while still forming a heterodimer with CLOCK (Hasegawa et al., 2019). We identified the same arginine at position 88 in zebrafish _bmal1a_ and mutated the codon to encode alanine (_UAS:R88A-DN-bmal1a_, Fig. S4A). Crossing the _UAS:R88A-DN-bmal1a_ with _kdrl:GAL4_ zebrafish adults yielded expression of _bmal1a_ throughout the whole vascular system in double-positive embryos (Fig. S4B), phenocopying the _kdrl:GAL4;UAS:DN-bmal1a_ line. Again, no gross morphological alterations were apparent and aortic vascular specification was unaffected, as indicated by normal _dll4_ expression at 28 hpf (Fig. S4C).

We next investigated whether HSPC numbers were altered in _kdrl:GAL4;UAS:DN-bmal1a_ embryos. We found no differences in _runx1_ expression at 28 hpf (Fig. S5A), or _cmyb_ (_myb_) expression at 36 hpf (Fig. S5B), indicating that HSPC specification was unaffected despite the expression of _DN-bmal1a_ in all endothelial cells, showing that Bmal1a exerts no cell-autonomous role on HSPC emergence. We also found no differences in _cmyb_+ cell numbers in the CHT region at 48 (Fig. S5C) and 54 hpf (Fig. S5D), indicating that HSPCs were able to colonise the CHT normally, at least at these timepoints. However, at 60 hpf, there was an increase in the number of _cmyb_+ cells in the CHT of double transgenic embryos (Fig. 2A), which extended to larvae at 4.5 dpf (Fig. 2B), as determined by quantifying the _cmyb_+ signal area. Similar observations were made when we used the second DN construct

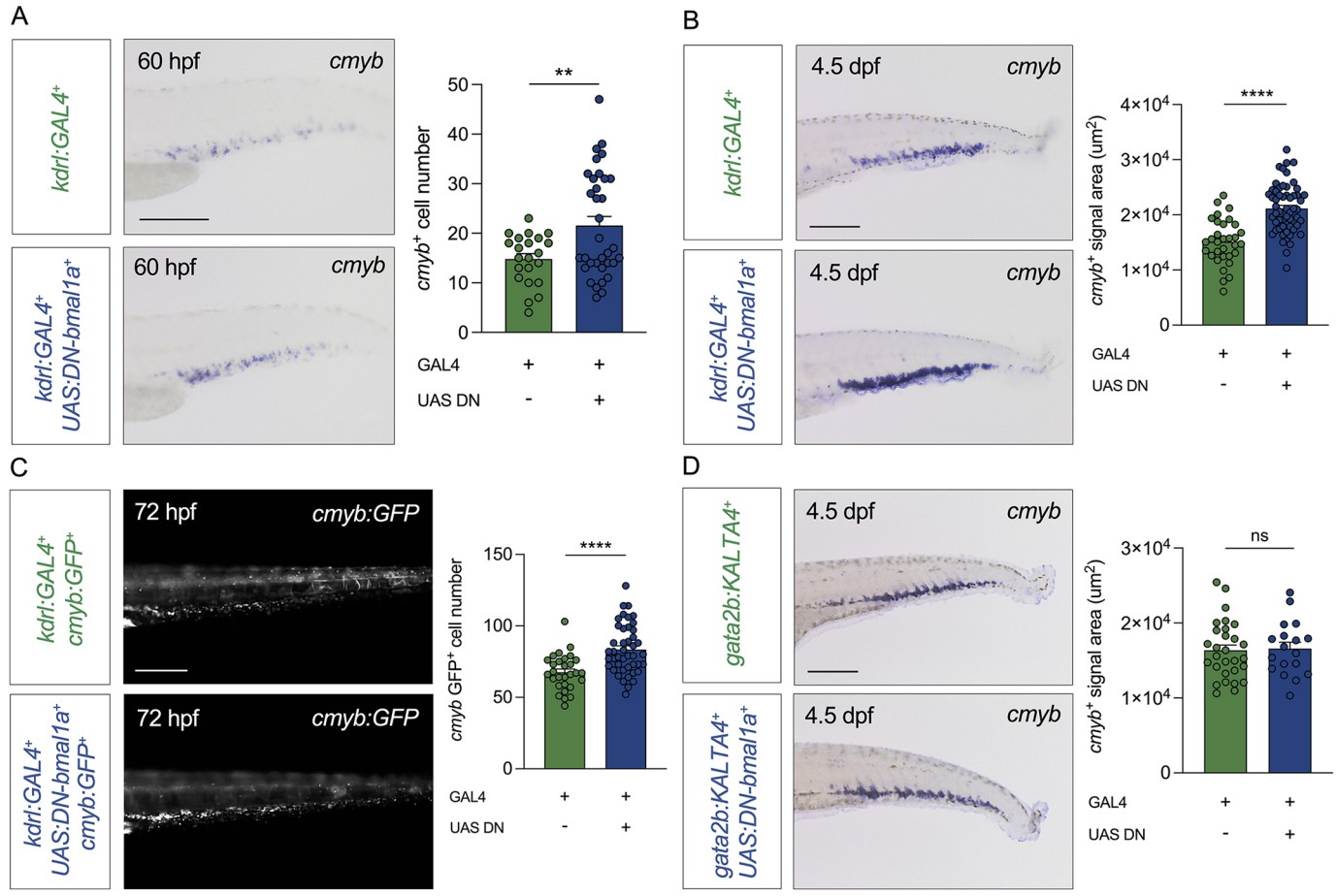

**Fig. 2. Endothelial-specific dominant-negative *bmal1a* results in a non-cell autonomous increase in haematopoietic stem and progenitor cell numbers in the caudal haematopoietic tissue.** (A) *cmyb in situ* hybridisation and quantification in *kdrl:GAL4;UAS:DN-bmal1a* embryos and controls at 60 hpf. (B) *cmyb in situ* hybridisation and quantification in *kdrl:GAL4;UAS:DN-bmal1a* larvae and controls at 4.5 dpf. (C) *cmyb:GFP⁺* cells and quantification in *kdrl:GAL4;UAS:DN-bmal1a;cmyb:GFP* larvae and controls at 72 hpf. (D) *cmyb in situ* hybridisation and quantification in *gata2b:KALTA4;UAS:DN-bmal1a* larvae and controls at 4.5 dpf. Statistical significance of the differences between the two groups was calculated using unpaired two-tailed Student's *t*-tests assuming equal variance (ns, not significant; **P<0.01, ****P<0.0001). Scale bars: 200 μm.

(*UAS:R88A-DN-bmal1a*), with no difference in *runx1* expression at 28 hpf (Fig. S4D) but an increase in *cmyb* expression in larvae at 4.5 dpf (Fig. S4E). Since both our dominant-negative *bmal1a* zebrafish lines gave identical phenotypes, we chose to continue with the *UAS:DN-bmal1a* line only, throughout the rest of the study.

The increase in HSPC numbers in the CHT was corroborated by using transgenic lines that fluorescently mark HSPCs. These lines (*cmyb:GFP*, *cd41:EGFP* and *runx1:NLS-mCherry*) were crossed with *kdrl:GAL4;UAS:DN-bmal1a* double-transgenic lines, and the resulting triple-transgenic larvae were analysed by microscopy. HSPC numbers were found to be significantly increased at 72 hpf in the CHT of triple-transgenic combinations relative to controls (Fig. 2C, Figs S6A,B). The increase in HSPCs remained present at 7 dpf, as determined by *cmyb* WISH (Fig. S6C).

In order to exclude any cell-autonomous effect of our *DN-bmal1a*, we placed the construct under the control of *gata2b:KALTA4*, in order to drive *DN-bmal1a* expression specifically in HSPCs (Butko et al., 2015). We found no difference in CHT-HSPC numbers in double-transgenic larvae at 4.5 dpf (Fig. 2D). Finally, we investigated whether the EC-specific dominant-negative Bmal1a had an effect on the number of differentiated cells in the CHT. We found no difference in the numbers of neutrophils (as determined by *mpx*) (Fig. S7A), erythrocytes (*gata1*) (Fig. S7B), macrophages

(*mfap4*) (Fig. S7C) or T-cells (*rag1*) (Fig. S7D) between *kdrl:GAL4;UAS:DN-bmal1a* larvae and controls at 4.5 dpf. This lack of difference in the number of differentiated cells in *kdrl:GAL4;UAS:DN-bmal1a* larvae may be attributed to the fact that the expanded HSPCs differentiate later or do not have the same differentiation potential, or that their expansion rate exceeds the rate of differentiation. Nevertheless, these findings demonstrate a role of CHT-EC-specific *bmal1a* in controlling HSPC expansion in a non-cell autonomous manner.

### Endothelial-specific dominant-negative Bmal1a results in an increased HSPC proliferation rate

Vascular morphology, such as expanded vascular plexus CHT, has a direct impact on the number of HSPCs (Xue et al., 2017; Tamplin et al., 2015). Furthermore, many factors derived from the CHT vasculature are known to positively regulate HSPC expansion (Wattrus and Zon, 2018). Thus, we reasoned that vascular morphology or the numbers of ECs in the CHT may be altered in *kdrl:GAL4;UAS:DN-bmal1a* embryos, resulting in the observed HSPC increase. However, we found no gross vascular morphological alterations in *kdrl:GAL4;UAS:DN-bmal1a;kdrl:EGFP* embryos (Fig. S3E) and the number of CHT-ECs between *kdrl:GAL4;UAS:DN-bmal1a;kdrl:nls-EGFP* zebrafish and *kdrl:*

*GAL4;kdrl:nls-EGFP* controls was similar at 48 hpf (Fig. S3F), when some HSPCs in the CHT may still be EGFP positive after specification from the dorsal aorta endothelium, but also at 4.5 dpf (Fig. S3G), when this is likely no longer the case.

We speculated that the increase in CHT-resident HSPCs in *kdrl: GAL4;UAS:DN-bmal1a* larvae may be due to a reduced migration rate of these cells away from this niche to the kidney glomerulus, which represents the adult niche. To assess this, we performed *cmyb* WISH in *kdrl:GAL4;UAS:DN-bmal1a* larvae and controls at 4.5 and 7 dpf, before imaging and quantifying *cmyb* signal in the kidney glomeruli. At 4.5 dpf, there was reduced *cmyb* WISH signal in the kidney glomeruli in *kdrl:GAL4;UAS:DN-bmal1a* larvae (Fig. S8A), while at 7 dpf this difference was no longer present (Fig. S8B). Together, these data suggest that there is a transient

delay in migration of HSPCs away from the CHT in the absence of functional endothelial Bmal1a.

We reasoned that the increase in HSPCs in the niche may also, at least in part, be due to a difference in their expansion rate in the CHT of *kdrl:GAL4;UAS:DN-bmal1a* animals. Therefore, we assessed the proliferation rate of CHT-HSPCs by staining for anti-phospho-histone 3 (PH3), a marker of mitotic cells (Hendzel et al., 1997), in *kdrl:GAL4;UAS:DN-bmal1a;cmyb:GFP* larvae and controls at 60 and 72 hpf. Triple-transgenic larvae showed a significant increase in the number of GFP$^+$PH3$^+$ cells in the CHT compared to controls at both 60 hpf (Fig. 3A) and 72 hpf (Fig. 3B). This indicated that the lack of functional Bmal1a in CHT-ECs results in an enhanced HSPC proliferation rate in the niche, which we decided to investigate further.

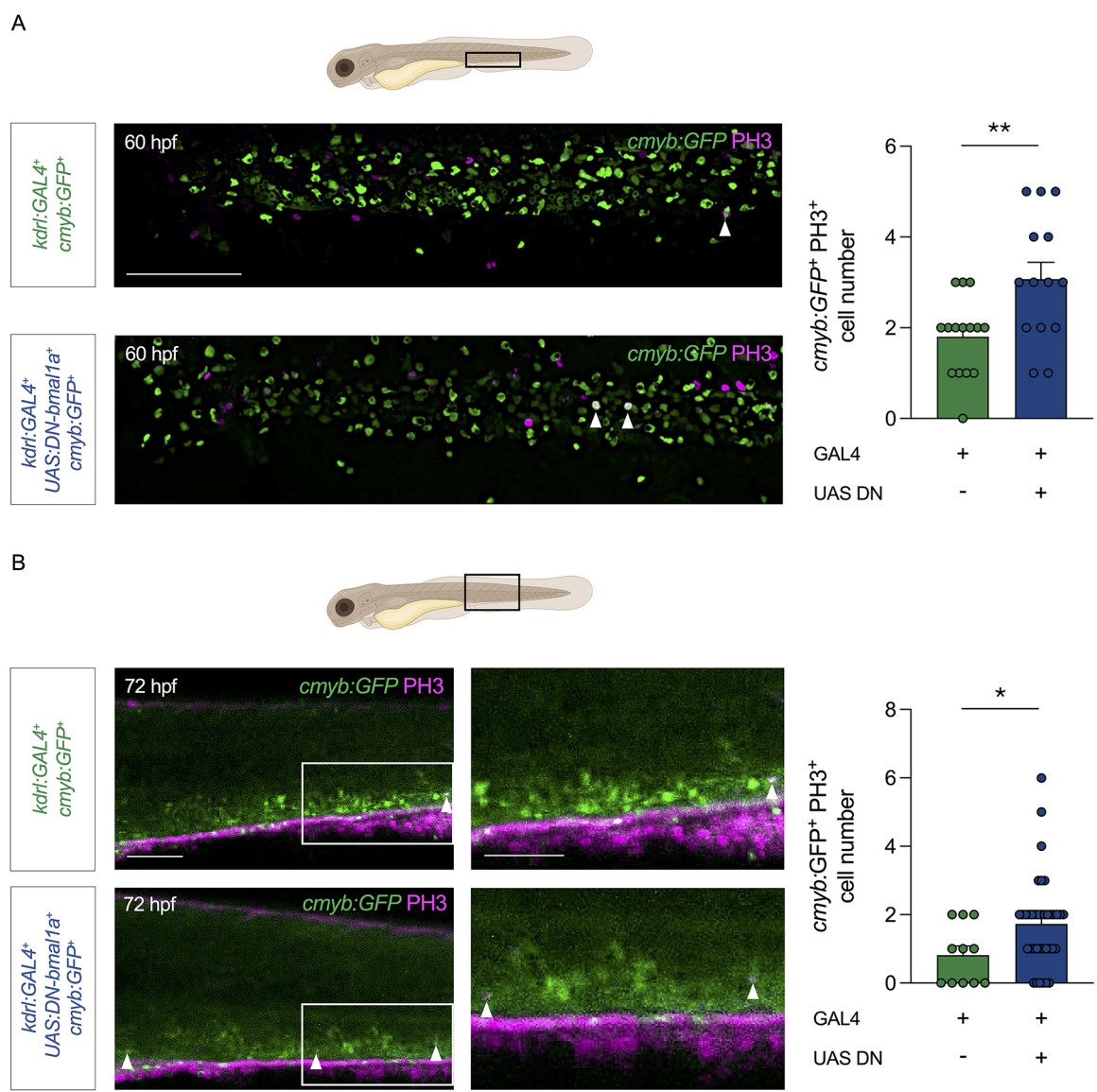

**Fig. 3. Endothelial-specific dominant-negative *bmal1a* embryos display an increased haematopoietic stem and progenitor cell proliferation rate in the caudal haematopoietic tissue.** (A) Immunohistochemistry and quantification of *cmyb:GFP* and phospho-histone 3 (PH3) double-positive haematopoietic stem and progenitor cells (HSPCs) in the caudal haematopoietic tissue (CHT) of *kdrl:GAL4;UAS:DN-bmal1a;cmyb:GFP* zebrafish larvae and controls at 60 hpf. (B) Immunohistochemistry and quantification of *cmyb:GFP* and PH3 double-positive HSPCs in the CHT of *kdrl:GAL4;UAS:DN-bmal1a;cmyb:GFP* zebrafish larvae and controls at 72 hpf. Statistical significance of the differences between two groups was calculated using an unpaired two-tailed Student's *t*-tests assuming equal variance (*$P<0.05$, **$P<0.01$). Schematic diagrams of embryos in A and B indicate the CHT regions imaged in each respective panel. Arrowheads indicate *cmyb:GFP* and PH3 double-positive cells. Scale bars: 200 μm (A and B, left); 50 μm (B, right). Created in BioRender by Petzold, T. (2026) https://BioRender.com/i9wqnnx. This figure was sublicensed under CC-BY 4.0 terms.

## Endothelial-derived glutamine fuels definitive HSPC proliferation

We hypothesised that the lack of functional Bmal1a in CHT-ECs results in alterations in transcriptional output, leading to enhanced HSPC proliferation. To investigate this, we dissected tails of *kdrl:GAL4;UAS:DN-bmal1a;kdrl:EGFP* embryos and of *UAS:DN-bmal1a;kdrl:EGFP* controls at 36 hpf, and used FACS to isolate EGFP$^+$ cells (Fig. 4A; for sorting strategy, see Fig. S9A). We subsequently performed bulk RNA-sequencing (RNA-seq) on these ECs and found 95 differentially regulated genes between the two genotypes (Fig. 4B,C). The majority of differentially regulated genes were found to be downregulated in *kdrl:GAL4;UAS:DN-bmal1a;kdrl:EGFP* embryos (84/95). A known direct target gene of Bmal1a, *nr1d2a* (Cho et al., 2012; Amaral and Johnston, 2012), was markedly downregulated in our RNA-seq dataset (Fig. 4B,C), providing further evidence of disrupted circadian clock function induced by the DN-Bmal1a protein.

The RNA-seq data revealed that *glud1a* was downregulated in *kdrl:GAL4;UAS:DN-bmal1a;kdrl:EGFP* embryos (Fig. 4B,C), which was confirmed by qPCR (Fig. S10A) and WISH (Fig. S10B). Glutamate dehydrogenase 1A (GLUD1A or GDH1), is a hexameric enzyme that catalyses the conversion of glutamate to α-ketoglutarate (Plaitakis et al., 2017; Shang et al., 2020). It counteracts the activity of GLUL (glutamate ammonia ligase), a second enzyme (glutamine synthetase) that produces glutamine *de novo* from glutamate and ammonia (Fig. 4D) (Eelen et al., 2018).

We reasoned that a downregulation of *glud1a* in our endothelial-specific *DN-bmal1a* zebrafish embryos could result in an accumulation of glutamate in ECs. This increase in the amount of available glutamate would ultimately result in an increase in glutamine production, due to an enhanced conversion of glutamate to glutamine by GLUL. To test this, we first examined the expression patterns of *glud1a* and *glul* in zebrafish by *in situ* hybridisation, to determine whether these genes are expressed in the CHT during the time of HSPC expansion. *glud1a* was found to be expressed in the CHT between 24 and 48 hpf (Fig. 4E), while *glula* (zebrafish possess three paralogues of the Glul gene family: *glula*, *glulb* and *glulc*) was expressed in the CHT between 24 and 60 hpf (Fig. 4E). Therefore, during and following aorta-derived HSPC colonisation of the CHT, *glula* may drive the production of glutamine in the niche. To investigate this hypothesis further, we quantified the amount of glutamine in the tails of 72 hpf *kdrl:GAL4; UAS:DN-bmal1a* larvae relative to controls. We observed a significant increase in the concentration of glutamine present in the EC-specific *DN-bmal1a* embryos in comparison to controls (Fig. 5A), suggesting that a reduction of Bmal1a function, and the resulting downregulation in *glud1a* expression, may ultimately account for an increased synthesis of glutamine.

Since glutamine is known to be necessary for the proliferation of many cell types, including malignant (Zhang et al., 2017; Nguyen and Duran, 2018) and stem cells (Shang et al., 2020; Yu et al., 2019), we hypothesised that the glutamine increase in the EC-specific *DN-bmal1a* embryos was the driver of the HSPC expansion phenotype observed in these animals. To investigate this, we supplemented wild-type zebrafish with 1 mM glutamine, following HSPC specification, between 48 hpf and 4.5 dpf. Glutamine supplementation resulted in increased HSPC numbers in the CHT, as determined by WISH (Fig. 5D), in line with previous *in vitro* data demonstrating the importance of glutamine for HSPC expansion (Ni et al., 2019). Together, these data suggests that the increase in glutamine production in *kdrl:GAL4;UAS:DN-bmal1a* embryos provides an explanation for the increased number of HSPCs in the CHT of these embryos.

We next investigated whether modulating GLUD1A activity affects HSPC numbers. We supplemented *kdrl:GAL4;UAS:DN-bmal1a* or wild-type zebrafish with the GLUD1A allosteric activators BCH (Han et al., 2016; Sener et al., 1981) and ADP (Li et al., 2011; Koberstein and Sund, 1973) between 48 hpf and 4.5 dpf. Supplementation of *kdrl:GAL4;UAS:DN-bmal1a* embryos with 500 µM BCH (Fig. 5B) or 100 µM ADP (Fig. 5C) resulted in decreased CHT *cmyb* signal at 4.5 dpf. Similarly, supplementation of wild-type embryos with either 500 µM BCH (Fig. 5E) or 100 µM ADP (Fig. 5F) also resulted in decreased CHT *cmyb* signal at 4.5 dpf. Wild-type embryos were then also supplemented with another allosteric activator of GLUD1A, L-leucine (Sener and Malaisse, 1980; Couee and Tipton, 1989). Supplementation of wild-type embryos with 1 mM L-leucine (Fig. 5G) also resulted in decreased CHT *cmyb* signal at 4.5 dpf.

Since the HSPC pool is very heterogeneous, we next wanted to understand which type(s) of progenitors are affected by GLUD1A activity and glutamine. To investigate this, we first treated *mindbomb* embryos with glutamine. *mindbomb* mutants are deficient for Notch signalling, and therefore lack all definitive HSPCs (Burns et al., 2005), but still produce erythromyeloid progenitors (Bertrand et al., 2010b). When treating *mindbomb* siblings with glutamine from 48 to 72 hpf, we found an increase in HSPC expansion, as determined by an increase in *cmyb* WISH signal at 72 hpf (Fig. S11A). *mindbomb*$^{-/-}$ embryos lacked *cmyb* signal in the CHT (Fig. S11B) and did not exhibit any increase following treatment with glutamine (Fig. S11B), demonstrating that only Notch-dependent aorta-derived HSPCs are sensitive to glutamine addition. To corroborate this finding, we supplemented *gata2b:KALTA4$^+$;UAS:lifeact-GFP$^+$* embryos with glutamine or L-leucine between 48 and 72 hpf. In line with our data using other HSPC reporter lines, we found more *gata2b:GFP$^+$* HSPCs in the CHT of glutamine-treated embryos at 72 hpf (Fig. S11C). *gata2b: KALTA4$^+$;UAS:lifeact-GFP$^+$* embryos treated with L-leucine had fewer *gata2b:GFP$^+$* cells in the CHT at 72 hpf (Fig. S11C), providing evidence that glutamine is important for the expansion of aorta-derived definitive HSPCs. Together, these results show that the *bmal1a-glud1a*-glutamine pathway can directly alter definitive (aorta-derived) HSPC expansion in the CHT, by modulating the amount of glutamine available to HSPCs.

## Glutamine is transported by SLC channels

Amino acids such as glutamine induce cell proliferation by activating the mTOR pathway (Jewell et al., 2015; Gonzalez et al., 2020). Glutamine is primarily transported via Slc (solute carrier)-type channels (Bhutia and Ganapathy, 2016; Yoo et al., 2020), some of which are able to carry out the export of glutamine, while others facilitate glutamine import. A further subset can both export and import glutamine (Bhutia and Ganapathy, 2016). We thus analysed the expression of Slc genes (pre-selected on their known capacity to transport glutamine) in CHT-ECs and HSPCs at 48 hpf, prior to the HSPC expansion phase. *kdrl:EGFP$^+$* ECs and *ikaros:EGFP$^{low}$* HSPCs (Mahony et al., 2018) were sorted by FACS (for sorting strategies, see Fig. S9A,B) from 48 hpf embryos and qPCR was performed to determine the expression of selected Slc genes in the sorted populations. *slc1a5* was determined to be the most highly expressed in CHT-ECs (Fig. S12A), while some other Slc genes, such as *slc38a5a* and *slc38a5b*, were also detected, albeit at a lower level. In the HSPC population, expression of importing Slc genes was analysed, as well as the expression of Slc genes encoding transporters that carry out glutamine import and export. As in ECs, *slc1a5* was determined to be the most highly expressed of the Slc genes examined in HSPCs, along with some expression of

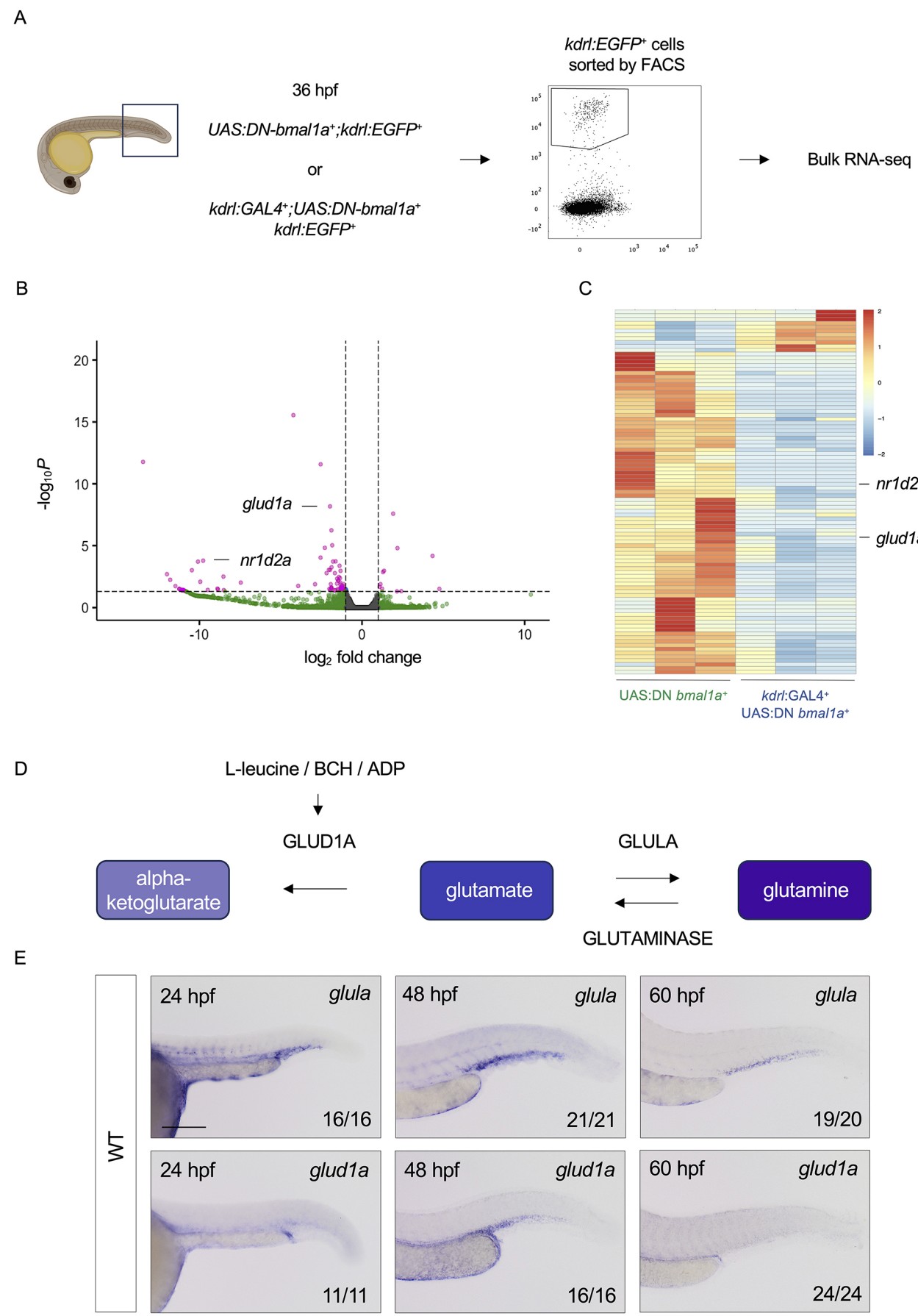

**Fig. 4.** See next page for legend.

**Fig. 4. Endothelial-specific dominant-negative *bmal1a* results in a decrease in *glud1a* expression in caudal haematopoietic tissue endothelial cells.** (A) Endothelial cells (ECs) were isolated by fluorescence-activated cell sorting (FACS) from *kdrl:GAL4;UAS:DN-bmal1a;cmyb:GFP* zebrafish embryos and controls at 36 hpf, before HSPC emergence and migration to the caudal haematopoietic tissue (CHT), and subjected to bulk RNA-sequencing; $n \approx 60$ embryos in triplicate for each genotype. (B) Volcano plot depicting the 95 genes that were differentially expressed in tail endothelial cells of *kdrl:GAL4;UAS:DN-bmal1a;cmyb:GFP* embryos and controls, with *glud1a* and *nr1d2a* indicated. (C) Heatmap depicting the 95 genes differentially expressed genes with *glud1a* and *nr1d2a* indicated. (D) Schematic depicting the conversion of glutamate to α-ketoglutarate and glutamine via the enzymes GLUD1A, GLULA and glutaminase. Allosteric activators of GLUD1A (L-leucine, BCH and ADP) are also depicted. (E) *In situ* hybridisation of *glula* and *glud1a* at 24, 48 and 60 hpf in wild-type embryos. The two genes are expressed in the CHT at times compatible with a role in developmental haematopoiesis. Scale bar: 200 µm. Created in BioRender by Petzold, T. (2026) https://BioRender.com/y7gmtpg. This figure was sublicensed under CC-BY 4.0 terms.

*slc38a5a*, *slc38a5b* and *slc38a2* (Fig. S12B). Together, the data demonstrate that genes encoding channels required for the transport of glutamine out of ECs and into HSPCs are expressed in the CHT at the beginning of HSPC expansion.

To probe whether SLC transporters do indeed play a role in HSPC expansion in the CHT, we treated wild-type embryos with the SLC blocker L-γ-glutamyl-*p*-nitroanilide (GPNA). GPNA was originally thought to be a specific Slc1a5 channel inhibitor (Esslinger et al., 2005), but has recently been shown to also block the activity of other sodium-dependent SLCs (Bröer et al., 2016) and also system L amino acid transporters, such as LAT1 and LAT2 (Chiu et al., 2017). Treating wild-type embryos with GPNA from 48 hpf to 4.5 dpf resulted in a complete loss of HSPCs in the CHT, as determined by *cmyb* WISH (Fig. S12F), indicating that glutamine transport into HSPCs may indeed be crucial not only for HSPC expansion but also for their survival.

### Loss of fetal liver endothelial *Bmal1* does not affect HSPC numbers in the mouse

Finally, we wanted to determine whether the HSPC expansion phenotype observed in the absence of functional EC-Bmal1a in zebrafish is conserved in mammals. To investigate this, we used a tamoxifen-inducible EC-specific mouse *Bmal1* KO line: *Cdh5:Cre^ERT2^;Bmal1^flox/flox^*. Pregnant mice were injected interperitoneally (i.p.) with 1 mg tamoxifen at embryonic day (E)10.5, following HSPC specification (Fig. S13A). Subsequently, embryo FLs were harvested at E13.5 and analysed by flow cytometry. ECs, as well as LSKs (lineage-negative Sca-1^+^ c-Kit^+^ HSPCs), were also isolated by FACS for gene expression analyses by qPCR (for the FACS gating strategy, see Fig. S14) and the purity of these populations was confirmed by qPCR analysis of *Cdh5* (Fig. S15A) and *Cmyb* (Fig. S15B) expression. *Bmal1* expression was reduced in FACS-sorted *Cdh5:Cre^ERT2^;Bmal1^flox/flox^* FL-ECs relative to controls (Fig. S13B), but this was not the case in sorted FL-LSKs (Fig. S13C), demonstrating EC-specific *Bmal1* deficiency.

Next, we assessed the impact of the EC-specific *Bmal1* KO on FL LSK numbers in mouse embryos at E13.5. We did not observe differences in LSK numbers in *Cdh5:Cre^ERT2^;Bmal1^flox/flox^* embryos compared to controls (Fig. S15C,D). Furthermore, we found that expression of the cell proliferation markers, *Mki67* (Fig. S15E) and *Ccnb2* (cyclin B2) (Fig. S15F) were unchanged in FL-LSKs between the two genotypes. Together, these results suggest that the HSPC expansion phenotype observed in the zebrafish CHT is not conserved in the analogous region in the mouse embryo: the FL.

We focused on investigating whether the *Glud1*-glutamate-*Glul*-glutamine pathway was affected in EC-specific *Bmal1* KO mouse embryos. *Glud1* expression was quantified in sorted FL-ECs by qPCR. *Glud1* expression was unchanged in *Cdh5:Cre^ERT2^;Bmal1^flox/flox^* ECs, relative to controls (Fig. S13D). This lack of transcriptional control of *Glud1* by Bmal1 in mouse FL-ECs provides an explanation as to why the HSPC expansion phenotype present in *kdrl:GAL4;UAS:DN-bmal1a* zebrafish was not recapitulated in *Cdh5:Cre^ERT2^;Bmal1^flox/flox^* mouse embryos. Furthermore, when analysing previously published single-cell transcriptomic data of human FL cells (Popescu et al., 2019), including ECs (Fig. S16A) and hepatocytes (Fig. S16B), we found that *Glud1* is in fact specifically expressed by hepatocytes but not ECs (Fig. S16C).

### The *Glud1*-glutamate-*Glul*-glutamine signalling axis is conserved in the mouse fetal liver

We next investigated whether the *Glud1*-glutamate-*Glul*-glutamine pathway is nevertheless conserved in the mouse FL. To assess this, we set up FL organ cultures in a glutamine-free medium before supplementing these with either glutamine or L-leucine, which is the allosteric activator of Glud1. After 48 h in culture, FL pieces in each condition were pooled (see Materials and Methods for details), and the resulting cell suspension was analysed by cytometry to measure the number of LSKs (Fig. 6A). All samples were fully acquired to determine the cell viability and total cell counts. Supplementation of FL organ cultures with glutamine or L-leucine did not affect cell viability (Fig. 6B) or total cell numbers (Fig. 6C) relative to glutamine-free controls. While the addition of glutamine did not alter the total number of LSKs (Fig. 6D) or their percentages (Fig. 6E), supplementation with L-leucine decreased the number of LSKs in organ cultures (Fig. 6D,E), similar to what was observed in zebrafish. Therefore, the importance of the *Glud1*-glutamate-*Glul*-glutamine axis in regulating HSPC expansion appears to be conserved in the mouse FL niche but is likely to be active in hepatocytes, rather than in ECs.

### DISCUSSION

A complete identification of genetic factors that govern HSPC expansion remains incomplete. Here, we have discovered a previously unreported mechanism by which HSPC expansion is controlled in the haematopoietic niche through regulation of the quantity of glutamine available for HSPCs from the niche microenvironment. We have demonstrated that the precise control of bioavailable glutamine is, at least in part, carried out by the core circadian clock component *bmal1a* in zebrafish, through regulation of *glud1a* expression. Our data also suggest that Bmal1a promotes the migration of HSPCs away from the CHT. Whether Bmal1a also plays a role in the regulation of HSPC immigration into the CHT in zebrafish will be interesting to investigate in future work. Furthermore, subsequent investigations using high-resolution 3D imaging of HSPCs in the CHT, kidney glomeruli and thymus may also enhance our understanding of HSPCs in these structures in the context of EC Bmal1a loss of function. Nevertheless, this study contributes to an expanding body of work implicating circadian clock genes not only in metabolic control (Reinke and Asher, 2019; Marcheva et al., 2013), but also in stem cell biology (Weger et al., 2017; Dierickx et al., 2018) and haematopoiesis (Golan et al., 2019; Mendez-Ferrer et al., 2009).

While previous studies focused on mechanisms that regulate HSPCs in a cell-autonomous manner, here we demonstrate that Bmal1 in zebrafish exerts its control on HSPC expansion non-cell autonomously. As such, along with previously reported transcription factors that control the HSPCs in CHT, such as Tfec

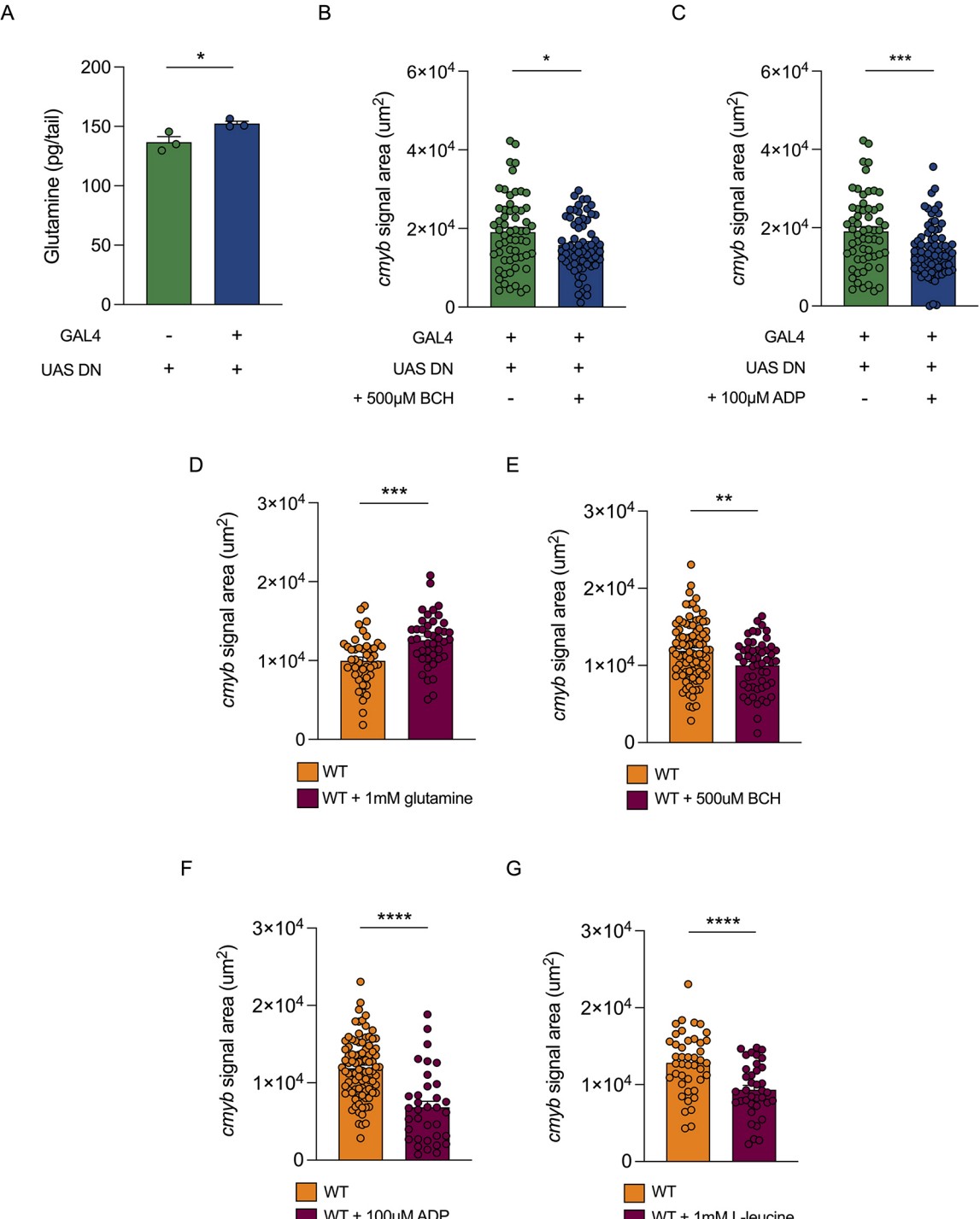

**Fig. 5. Glutamine concentration is increased in endothelial-specific dominant-negative _bmal1a_ zebrafish embryo tails and results in increased haematopoietic stem and progenitor cell numbers.** (A) Quantification of glutamine concentrations in dissected tails of _kdrl:GAL4;UAS:DN-bmal1a_ larvae and controls. (B) Quantification of _cmyb in situ_ hybridisation signal in _kdrl:GAL4;UAS:DN-bmal1a_ larvae treated with 500 µM BCH and controls at 4.5 dpf. (C) Quantification of _cmyb in situ_ hybridisation signal in _kdrl:GAL4;UAS:DN-bmal1a_ larvae treated with 100 µM ADP and controls at 4.5 dpf. (D) Quantification of _cmyb in situ_ hybridisation signal in wild-type larvae treated with 1 mM glutamine and controls at 4.5 dpf. (E) Quantification of _cmyb in situ_ hybridisation signal in wild-type larvae treated with 500 µM BCH and controls at 4.5 dpf. (F) Quantification of _cmyb in situ_ hybridisation signal in wild-type larvae treated with 100 µM ADP and controls at 4.5 dpf. (G) Quantification of _cmyb in situ_ hybridisation signal in wild-type larvae treated with 1 mM L-leucine and controls at 4.5 dpf. Statistical significance of the differences between two groups was calculated using unpaired two-tailed Student's _t_-tests assuming equal variance (*$P<0.05$, **$P<0.01$, ***$P<0.001$, ****$P<0.0001$).

(Mahony et al., 2016) and Klf6a (Xue et al., 2017), we demonstrate that Bmal1a is a newly identified transcriptional regulator of HSPCs in the niche. Puram et al. previously showed that human HSPCs

possess robust rhythmic expression of _Per2_, which is a direct target gene of BMAL1 (Puram et al., 2016). Contrary to this, our data in the zebrafish demonstrate that _bmal1a_ expression in CHT-ECs does

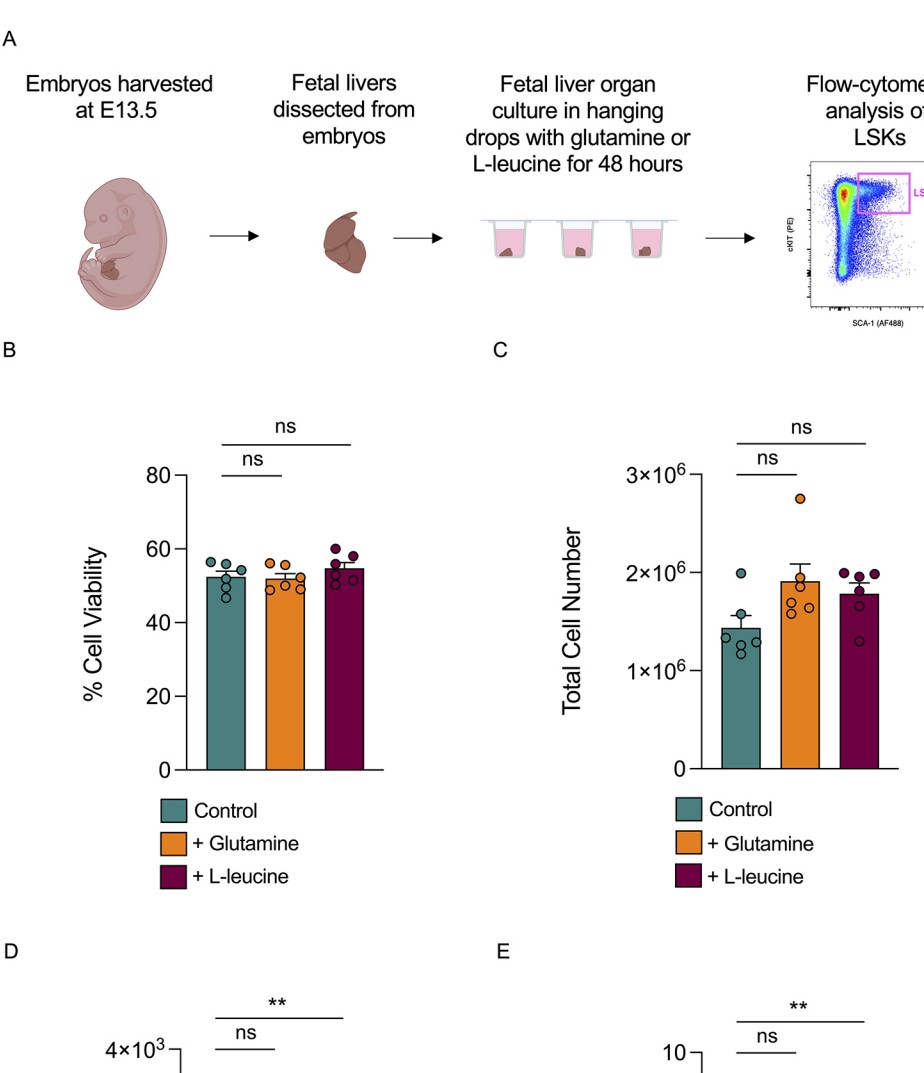

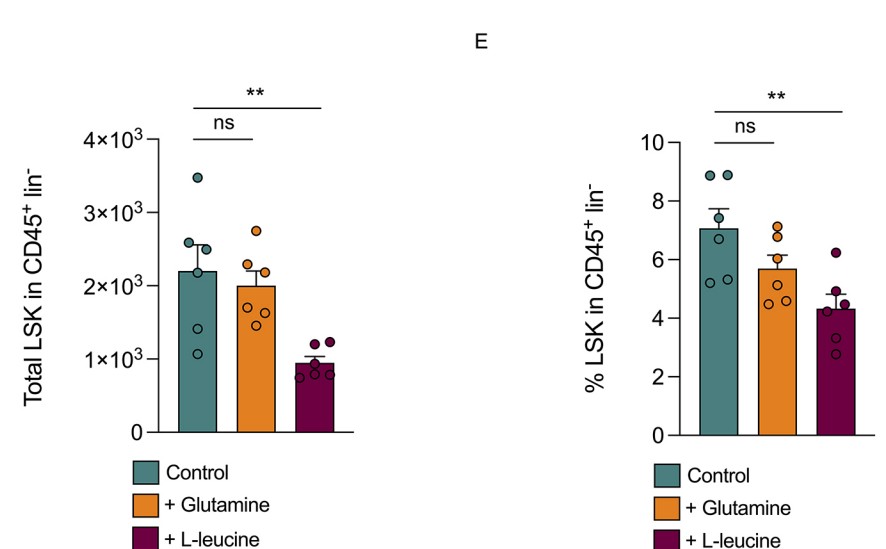

**Fig. 6. Increasing GLUD1 activity in mouse fetal liver organ culture results in a decrease in LSKs (lineage-negative Sca-1$^+$ c-Kit$^+$ haematopoietic stem cells).** (A) Schematic of fetal liver organ culture experiments. Dissected fetal liver pieces were cultured with glutamine or L-leucine, or without either (controls) for 48 h prior to flow cytometry analysis. (B) Viability (percentage of live cells) of fetal liver organ culture cells in controls, or when supplemented with glutamine (2 mM or 10 mM, see Materials and Methods) or 10 mM L-leucine. (C) Total fetal liver organ culture cell numbers in controls, or when supplemented with glutamine (2 mM or 10 mM) or 10 mM L-leucine. (D) Total LSKs in CD45$^+$ lin$^-$ cells in controls or when supplemented with glutamine (2 mM or 10 mM) or 10 mM L-leucine. (E) Percentage of LSKs in CD45$^+$ lin$^-$ cells in controls, or when supplemented with glutamine (2 mM or 10 mM) or 10 mM L-leucine. Statistical significance of the differences between two groups was calculated using unpaired two-tailed Student's $t$-tests assuming equal variance (ns, not significant; **$P<0.01$). Created in BioRender by Petzold, T. (2026) https://BioRender.com/59ytymc. This figure was sublicensed under CC-BY 4.0 terms.

not cycle and is present only at 24-42 hpf. As such, the role of Bmal1a in regulating CHT-resident HSPCs occurs in a non-rhythmic manner. Interestingly non-rhythmic circadian clock gene expression has previously also been shown to also occur in the fetal liver in rats (Varcoe et al., 2013; Wharfe et al., 2011), suggesting that the lack of rhythmic circadian clock gene expression in the embryonic HSPC expansion niche is conserved across species. Further research will be required to better understand the mechanisms that induce rhythmic circadian clock gene expression in some tissues but not others.

Our study indicates that the *bmal1a-glud1a-glula*-glutamine pathway functions as a previously unidentified metabolic crosstalk between CHT-ECs and HSPCs, in line with a recent study showing a similar mechanism to enhance satellite cell proliferation during muscle regeneration (Shang et al., 2020). In their study, Shang and colleagues showed that *Glud1*-knock-out macrophages synthesise more glutamine, which is then exported out of these cells before being taken up by satellite cells, boosting their expansion and subsequent muscle regeneration. Their study was caried out in the context of muscle damage, in which the tissue must regenerate, and a downregulation in *Glud1* to generate additional glutamine was beneficial. Similarly, our research demonstrates how expression of *bmal1a* and its downstream target *glud1a* are lost in CHT-ECs prior to the arrival of HSPCs in the niche. This downregulation in *glud1a*

expression then results in increased glutamine production at the time it is required by incoming HSPCs. We suggest that this *bmal1a-glud1a-glula*-glutamine axis, acting in a highly time-sensitive manner, represents a newly identified homeostatic control mechanism to precisely regulate HSPC numbers in the CHT. With this in mind, it is tempting to speculate that the mis-regulation of *glud1a* expression or protein activity may be a contributing factor in uncontrolled haematopoietic cell expansion in blood cancers, as leukaemia cells are known to be dependent on glutamine (Gregory et al., 2019; Sancerni et al., 2022).

Our Slc gene expression data in ECs and HSPCs in the CHT suggest that the encoded channels may facilitate the export of glutamine out of ECs, before allowing uptake of glutamine into HSPCs. We found *slc1a5* to be the most highly expressed glutamine export gene in CHT-ECs. Previous reports also showed high expression of *slc1a5* in both human (Oburoglu et al., 2014) and mouse (Ni et al., 2019) HSPCs. While SLC1A5 has been shown to be important for HSPC commitment to the erythroid lineage (Oburoglu et al., 2014), we find it also plays an important role in HSPC expansion. SLC1A5-mediated cellular import of neutral amino acids such as glutamine is known to stimulate mammalian target of rapamycin complex 1 (mTORC1) signalling (Nicklin et al., 2009; Bhutia et al., 2015), a well-known driver of cell growth and proliferation, by activating anabolic processes such as DNA and protein synthesis (Valvezan and Manning, 2019; Mossmann et al., 2018; Saxton and Sabatini, 2017). In our study, we used GPNA, which blocks SLC-mediated glutamine transport but also inhibits the uptake of other neutral amino acids (Bhutia and Ganapathy, 2016; Chiu et al., 2017; Fuchs and Bode, 2005), rendering it difficult to discern the relative contribution that a lack of glutamine export or uptake has on the phenotype observed. Recently however, Miklas et al. demonstrated that zebrafish and neonatal mouse hearts can regenerate through cardiomyocyte de-differentiation and proliferation, resulting from glutamine-driven mTORC1 activation (Miklas et al., 2022). Taken together, these previous findings and the data we present in this article suggest that mTORC1 is likely to be a key component of the molecular pathway by which EC-derived glutamine induces HSPC proliferation, although further research will be required to confirm this hypothesis.

The data generated from our EC-specific *Bmal1*-KO mouse embryo experiments demonstrate that *Glud1* expression is not regulated by Bmal1 in FL-ECs, providing an explanation as to why FL LSK numbers were unaltered in these animals in comparison to controls. However, our data indicate that the *Bmal1-Glud1-Glul*-glutamine pathway is likely conserved in the FL, but that it is not predominantly active in ECs as in the zebrafish CHT. Rather, single-cell transcriptome analyses of human FL cells suggests that this pathway is active in hepatocytes in mammals. Through evolution from teleosts, such as zebrafish, to mammals, HSPCs expand in a niche that has become increasingly more complex (Mahony and Bertrand, 2019): while the CHT in zebrafish is a transient vascularised tissue (Tamplin et al., 2015), the FL in mammals is a bona fide organ containing hepatocytes. Mouse FL hepatocytes have been shown to express *Bmal1* (Ceccacci et al., 2023), *Glud1* (Ceccacci et al., 2023) and *Glul* (Kuo et al., 1988), when glutamine is known to be present in, and exported from, hepatocytes (Gebhardt and Coffer, 2013; Watford, 2000). The cell-type expression differences between key regulators of the HSPC niche in zebrafish and mammals was recently highlighted in another study in which we reported that *ifi30* (*ifi30a*), a gene that plays an important role in promoting HSPC expansion by detoxifying the niche from ROS, is specifically expressed in zebrafish CHT-ECs, but is only expressed in macrophages in the human FL (Cacialli et al., 2021).

This, together with our findings in the present study, demonstrates how, although the HSPC niche has evolved in complexity over time, crucial genetic pathways, such as the *bmal1a-glud1a-glula*-glutamine axis, have remained conserved.

In this article, we have elucidated a molecular pathway through which Bmal1 reprograms glutamine metabolism in the HSPC niche of both zebrafish and mouse, by controlling genes involved in glutamine synthesis. This ultimately changes glutamine availability in the niche, affecting HSPC expansion. Our findings reveal a previously unreported, non-cell autonomous, homeostatic mechanism that controls embryonic HSPC proliferation. Our mouse data demonstrate the sensitivity of the LSK population (containing bona fide HSCs) in response to Glud1 activity modulation. The lack of robust markers to discriminate HSCs from other progenitor subsets in the zebrafish model means we were unable to tease apart which of the definitive HSPC subsets respond to glutamine; this will be interesting and important to clarify in the future. Manipulation of the *bmal1a-glud1a-glula*-glutamine axis identified in this work may lead to new strategies to precisely control HSPC proliferation *ex vivo*, which could pave the way for greatly improved regenerative medicine protocols.

## MATERIALS AND METHODS
### Ethical statement
All animals, zebrafish and mice, were raised in accordance to FELASA and Swiss guidelines (Alestrom et al., 2020). All animal procedures and experiments were performed in strict accordance with all mandatory guidelines (EU and Swiss directives on the protection of animals used for scientific purposes), and approved and carried out in accordance with the guidelines of the animal research committee of Geneva [Commission Cantonale pour les Expériences sur les Animaux (CCEA) and the Office fédéral de la santé alimentaire et des affaires vétérinaires (OSAV)], under licence GE/178/19. All efforts were made to comply to the 3R guidelines.

### Mouse models and timed matings
$Tg(Cdh5:Cre^{ERT2+/-};Bmal1^{flox/flox})$ (referred to as $Cdh5:Cre^{ERT2+/-};Bmal1^{flox/flox}$ throughout) mice were generated by crossing $Tg(Cdh5:Cre^{ERT2+/+})$ (referred to as $Cdh5:Cre^{ERT2}$ throughout) (B6) mice (a gift from R. Adams, Max Planck Institute for Molecular Biomedicine, Münster, Germany) with $Tg(Bmal1^{flox/flox})$ (referred to as $Bmal1^{flox/flox}$ throughout) (B6) mice (Jackson Laboratories) at ENVIGO. For timed matings, $Tg(Cdh5:Cre^{ERT2+/-};Bmal1^{flox/flox})$ (referred to as $Cdh5:Cre^{ERT2}:Bmal1^{flox/flox}$) mice were crossed with $Tg(Bmal1^{flox/flox})$ mice to generate $Cdh5:Cre^{ERT2}$-positive and -negative littermate controls. One female mouse was placed with a single male for 12 h overnight and subsequently examined for the presence of a copulation plug. This finding was used as an indicator of potential pregnancy, and marked as embryonic day (E) 0.5. At E10.5, pregnant females were intraperitoneally (i.p.) injected with 1 mg tamoxifen (prepared as described below). At E13.5, pregnant females were euthanised and the fetal livers of the embryos were harvested for FACS and qPCR analysis. A sample of the remaining embryonic tissue was also kept for genotyping of the embryos by PCR.

### Fetal liver organ cultures
For fetal liver organ cultures, E13.5 FLs were dissected from wild-type C57BL/6 pregnant mice, and the culture was performed as previously described (Bertrand et al., 2005, 2006). Briefly, each FL was cut into eight pieces, and each piece was placed into single wells of a 60-well Terasaki plate containing 30 µl of medium composed of RPMI (deprived of glutamine), 10% fetal calf serum, penicillin/streptomycin (10 U/ml), 0.1% β-mercaptoethanol and HEPES (25 mM). The plates were then inverted and placed in an incubator at 37°C for 48 h. Glutamine (Sigma-Aldrich) was added to produce a final medium concentration of either 2 mM or 10 mM. L-leucine (Sigma-Aldrich) was added to a final medium concentration of 10 mM. Each condition was tested on six different FLs and two independent

experiments were performed. At the end of the culture, all the eight pieces originating from the same FL were consolidated for FACS analysis. Since no differences were found for the readouts tested between the 2 mM or 10 mM glutamine conditions, data for these were collated.

### Tamoxifen preparation

Tamoxifen powder was purchased from Sigma (T5648-5G). The power was first dissolved in ethanol at 50 mg/ml, then diluted with castor oil to 25 mg/ml. Before injection, the tamoxifen solution was further diluted to 5 mg/ml with 1×phosphate-buffered saline (PBS). Pregnant females were i.p. injected with 200 µl of the 5 mg/ml tamoxifen solution at E10.5.

### Zebrafish husbandry

AB* zebrafish, as well as transgenic zebrafish lines were kept in a 14/10 h light/dark cycle at 28.5°C. Embryos were obtained as described previously (Westerfield, 2000). Embryos were staged by hours-post fertilisation (hpf) as described previously (Kimmel et al., 1995). In this study, the zebrafish *mindbomb*[*ta52b*] (Itoh et al., 2003) mutant was used, as well as the following transgenic zebrafish lines: *Tg(kdrl:GAL4)*[*bw9*] (Kim et al., 2014) (referred to as *kdrl:GAL4*), *Tg(gata2b:KALTA4)*[*sd32*] (Butko et al., 2015) (referred to as *gata2b:KALTA4*), *Tg(kdrl:EGFP)*[*s843*] (Jinn et al., 2005) (referred to as *kdrl: EGFP*), *Tg(kdrl:EGFP-NLS)* (Blum et al., 2008) (referred to as *kdrl:nls-EGFP*), *Tg(cmyb:GFP)*[*zf169*] (North et al., 2007) (referred to as *cmyb:GFP*), *Tg(Mmu.Runx1:NLS-mCherry)*[*cz2010*] (Tamplin et al., 2015) (referred to as *runx1:mCherry*), *Tg(CD41:EGFP)* (Lin et al., 2005) (referred to as *cd41: EGFP*), *Tg(ikaros:EGFP)*[*fr101*] (referred to as *ikaros:EGFP*) and *Tg(UAS: lifeact-GFP)*[*mu271*] (Butko et al., 2015). Zebrafish embryos were treated with 0.003% 1-phenyl-2-thiourea (PTU, Sigma P7629) starting at 24 hpf to prevent pigmentation.

### Generation of transgenic animals

For *Tg(UAS:DN-bmal1a)* and *Tg(UAS:R88A-DN-bmal1a)* zebrafish generation, a Tol2 vector containing 4xUAS promoter, the coding sequence (including a STOP codon) and a poly-adenylation signal sequence was generated by subcloning. The Tol2 *UAS:R88A-DN-bmal1a* construct was generated by first generating a *UAS:full-length bmal1a* construct, before carrying out site-directed mutagenesis of this using a QuikChange II kit (Agilent). AB* zebrafish embryos were co-injected with 50 pg of the final Tol2 vector along with 50 pg of *tol2 transposase* mRNA. Injected F0s were mated with AB* zebrafish, and the resulting F1 offspring were screened by PCR to assess germline integration of the Tol2 construct. Primers used for the generation of the *UAS:DN-bmal1a* and *UAS:R88A-DN-bmal1a* constructs, and for genotyping of UAS lines are listed in Tables S2 and S3, respectively.

### Whole-mount *in situ* hybridisation and analyses

Whole-mount *in situ* hybridisation was performed on 4% paraformaldehyde-fixed embryos, as described previously (Thisse and Thisse, 2008). Digoxigenin-labelled *runx1*, *cmyb*, *dll4*, *mpx*, *gata1*, *mfap4* (*mfap4.1*) and *rag1* probes were previously described (Mahony et al., 2016; Petzold et al., 2025). *bmal1a*, *bmal1b*, *clocka*, *glula*, *glud1a*, *slc1a5*, *slc38a5a* and *slc38a5b* digoxigenin-labelled probes were synthesised using an RNA labelling kit (SP6/T7; Roche), using primers listed in Table S1. RNA probes were generated by linearisation of TOPO-TA or ZeroBlunt vectors (Invitrogen) containing the PCR-amplified cDNA sequences.

### Immunofluorescence

Transgenic fluorescent embryos were embedded in 1% agarose in a glass-bottomed dish. Immunofluorescence double staining was performed as described previously (Gao et al., 2015), with chicken anti-GFP (1:400; Life Technologies, A10262) and rabbit anti-phospho-histone 3 (PH3) antibodies (1:250; Abcam, ab308373). AlexaFluor 488-conjugated anti-chicken secondary antibody (1:1000; Life Technologies, A11039) and AlexaFluor 594-conjugated anti-rabbit secondary antibody (1:1000; Life Technologies, A11012) were used to reveal the primary antibodies.

### Zebrafish chemical treatments

All compounds used in this study were purchased from Sigma-Aldrich. Zebrafish were treated with glutamine at a final concentration of 1 mM,

BCH at 500 µM, ADP at 100 µM, L-leucine at 1 mM and GPNA at 500 µM. Zebrafish were exposed to compounds in 0.003% 1-phenyl-2-thiourea (PTU, Sigma, P7629) E3 (fish) water in multiwell plates between 48 hpf and 4.5 dpf, unless stated otherwise. Compound water was replaced every 12 h. Following exposure, embryos were fixed in 4% paraformaldehyde.

### Glutamine assay

Tails of 72 hpf *kdrl:GAL4;UAS:DN-bmal1a* larvae and *kdrl:GAL4* controls were dissected. Three independent experiments were carried out using clutches of ~35 embryos per condition. The glutamine concentration per embryo tail was then quantified using a colorimetric glutamine assay kit following the manufacturer's protocol (Abcam, ab197011).

### Microscopy

Whole-mount *in situ* hybridisation images were taken on an Olympus MVX10 microscope in 100% glycerol. Fluorescent images were taken with an Olympus IX83 microscope. Representative fluorescent confocal images in Fig. 3A were taken using an upright 3i spinning-disc confocal microscope and a Zeiss Plan-Apochromat water-dipping objective. Representative fluorescent confocal images in Fig. 3B were taken using a Nikon inverted A1r spectral microscope. Fluorescent confocal images in Fig. 3A are 3D maximum projection images from z-stack acquisitions, while fluorescent confocal images in Fig. 3B were taken in 2D. All images were taken using the CellSens Dimension software (Olympus) apart from confocal images, which were taken using the NIS-Elements Advanced Research software (Nikon).

### Image processing and quantification

All images were processed using Fiji ImageJ (NIH) (Schindelin et al., 2012). *bmal1a*, *runx1* and *cmyb in situ* hybridisation signal intensity of the desired region was quantified as described previously (Dobrzycki et al., 2018). Quantification of *cmyb*-, *mpx*- and *mfap4*-positive cell numbers in the desired region in *in situ* hybridisation images was carried out manually using the counter tool following image inversion, sharpening and enlarging the region of interest of each image in Fiji ImageJ, as was carried out previously (Cacialli et al., 2021, 2022). *cmyb*, *gata1* and *rag1 in situ* hybridisation expression area in particular regions was measured manually in Fiji ImageJ, as has been carried out previously (Lundin et al., 2020; Soto et al., 2021; Klaus et al., 2022; Ghersi et al., 2023; Cacialli et al., 2021; Aleman et al., 2026). In all *in situ* hybridisation quantifications, aortic signals along the trunk to the end of the yolk tube extension, and CHT signals from the end of the yolk tube extension to the end of the tail were quantified, respectively. *kdrl:nls-EGFP*, *cmyb:GFP*-, *cd41:EGFP*[*low*]- and *runx1:nls-mCherry*-positive cell numbers in the CHT were quantified manually from 2D images using the counter tool following image inversion, sharpening and enlarging the region of interest of each image in Fiji ImageJ, as was carried out previously (Cacialli et al., 2021, 2022). Positive cells from the end of the yolk tube extension to the end of the tail were quantified.

### Fluorescence activated cell sorting

Whole-zebrafish embryos or tail dissections were incubated with a liberase-blendzyme 3 (Roche) solution for 90 min at 33°C, then dissociated and resuspended in 0.9× PBS/1% fetal calf serum, as described previously (Cacialli et al., 2021). We distinguished and excluded dead cells by staining them with SYTOX-red (Life Technologies) or DRAQ7 (Thermo Fischer Scientific). Cell sorting was performed using an Aria II (BD Biosciences, software diva v6.1.3) or BIORAD S3 cell sorter. Cell suspensions were passed through a 40 mm filter prior to FACS. Data were acquired on a LSR2Fortessa (BD Biosciences, software diva8.0.2) and analysed with FlowJo. Fetal livers were dissociated into 1% BSA in PBS by pipetting, and then underwent red blood cell lysis using RBC Lysis Buffer (Biolegend, 420302) for 5 min at room temperature. Cells were then washed and blocked with FcR Blocking Reagent (mouse; 1:50) from Miltenyi Biotec (130-092-75) for 15 min at room temperature. Cells were surface stained with CD31 (clone MEC13.3), Sca1 (clone D7), CD45 (clone 30-F11), cKit (CD117; clone2B8) and lineage (CD19, clone 6D5; GR1, clone RB6-8C5; TER119, clone TER-119; and CD3, clone 145-2C11) antibodies (1:100) for 15 min on ice. DAPI was used to distinguish live cells (2 µg per 1 million cells;

AppliChem, A4099). Cells were analysed and sorted using an Aria Fusion cell sorter (BD Biosciences) and flow cytometry data analysed using FlowJo software (BD). For qPCR, cells were directly sorted into Qiagen RLT lysis buffer (79216).

## Quantitative real-time PCR and analyses

Total RNA was extracted using RNeasy minikit (Qiagen) and reverse transcribed into cDNA using a Superscript III kit (Invitrogen). Quantitative real-time PCR (qPCR) was performed using a KAPA SYBR FAST Universal qPCR Kit (KAPA BIOSYSTEMS) and run on a CFX connect real-time system (Bio-Rad). All qPCR primers used for gene expression in zebrafish and mouse are listed in Table S4. All qPCR experiments were performed using technical triplicates. Experiments were each repeated three times and fold-change averages from each experiment were combined.

## Bulk RNA-sequencing and analyses

The total RNAs of sorted GPF$^+$ cells from the tails of 36 hpf *kdrl:EGFP* embryos were extracted using a QIAGEN RNeasy Mini Kit. The mRNA sequencing libraries were generated using a SMARTer Nextera kit for Illumina. The library preparations were sequenced on an Illumina Hiseq 4000 platform and 150 bp paired-end reads were generated. The fastq files were mapped to the UCSC Danio rerio danRer10 (GRCz10) genome with STAR v2.7.0f (Dobin et al., 2013). The biological QC was performed with picard tools. The number of reads mapping to each gene feature of the UCSC Danio rerio danRer10 reference was prepared with HTSeq v0.9.1 (HTseq-count) (Putri et al., 2022; Anders et al., 2015). The differential expression analysis was performed with the statistical analysis R/Bioconductor package edgeR 1.34.1 (Robinson et al., 2010), with a multiple testing Benjamini and Hochberg correction FDR 5% and a fold-change threshold of 2. Differentially expressed genes are listed in Table S5.

## Data analyses

Statistical significance between two samples was calculated using unpaired two-tailed Student's *t*-tests assuming unequal variance. At least three independent experiments were carried out in all cases, unless stated otherwise. In all experiments, normality was assumed and variance was comparable between groups. Sample size was selected empirically according to previous experience in the assessment of experimental variability. The investigators were blinded for all transgenic zebrafish and mouse flow cytometry experiments, both during the experiments and the quantification. Circadian rhythmicity in Fig. S1 was determined by cosinor analysis (Spadaro et al., 2020; Refinetti et al., 2007). Numerical data are the mean±s.e.m., unless stated otherwise. Statistical calculations and the graphs for the numerical data were performed using Prism 10 software (GraphPad Software). Details of the statistical analyses for the bulk RNA-seq experiment are provided in the corresponding section.

## Acknowledgements

We thank all lab members from the Neerman-Arbez, Scheiermann and Bertrand laboratories for their comments and suggestions during this project. We are also grateful to the animal facility staff for excellent zebrafish and mouse husbandry, to the flow cytometry facility for their assistance with FACS and flow cytometry, to the bioimaging facility for their help with confocal imaging, and to the genomics platform at UNIGE for performing the RNA-sequencing workflow.

## Competing interests

The authors declare no competing or financial interests.

## Author contributions

Conceptualization: T.P., L.K.L., K.N.I.B., R.G., C.S., J.Y.B; Data curation: T.P., L.K.L., K.N.I.B., A.K., B.B., S.J.; Formal analysis: T.P., L.K.L., K.N.I.B.; Funding acquisition: C.S., J.Y.B; Methodology: T.P., R.G., J.Y.B; Resources: H.G., C.S., J.Y.B.; Supervision: C.S., J.Y.B; Writing – original draft: T.P., J.Y.B; Writing – review & editing: T.P., L.K.L., C.S., J.Y.B.

## Funding

This work was supported by a grant from the Fondation Privée des Hopitaux de Genève (RC05-03 to J.Y.B. and C.S.). T.P. benefitted from a grant from the Gabbiani Fund. J.Y.B. was funded by the Schweizerischer Nationalfonds zur Förderung der Wissenschaftlichen Forschung (320030-228027). C.S.

was funded by the European Research Council (101001233, CIRCADYN), the Deutsche Forschungsgemeinschaft [TRR359 (491676693), project B07], the Schweizerischer Nationalfonds zur Förderung der Wissenschaftlichen Forschung (310030_182417/1), the Krebsliga Schweiz (KLS-4836-08-2019) and the Ligue Genevoise Contre le Cancer (2106). Open Access funding provided by the University of Geneva. Deposited in PMC for immediate release.

## Data and resource availability

Raw RNA-seq data will be deposited in Yareta (doi:10.26037/yareta:2l3zxli34zdd3pgeq4cst75q5u) and have been deposited in GEO under accession number GSE282622. All other relevant data and details of resources can be found within the article and its supplementary information.

## Peer review history

The peer review history is available online at https://journals.biologists.com/dev/lookup/doi/10.1242/dev.204726.reviewer-comments.pdf

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
