## [Peer Review File · Development (Cambridge, England)]

BMAL1 modulates glutamine supply to control haematopoietic stem and progenitor cell expansion

Tim Petzold, Lydia K. Lutes, Keila Navarro I. Batista, Antonia Konle, Bastien Baechler, Stéphane Jemelin, Holger Gerhardt, Rachel Golub, Christoph Scheiermann and Julien Y. Bertrand

DOI: 10.1242/dev.204726

Editor: Steve Wilson

Review timeline

Original submission:	18 February 2025
Editorial decision:	31 March 2025
First revision received:	30 September 2025
Editorial decision:	6 November 2025
Second revision received:	5 February 2026
Accepted:	11 March 2026

Original submission

First decision letter

MS ID#: dev.204726

MS TITLE: BMAL1 modulates glutamine supply to control hematopoietic stem and progenitor cell expansion

AUTHORS: Tim Petzold, Lydia K. Lutes, Keila Navarro I. Batista, Antonia Konle, Bastien Baechler, Stéphane Jemelin, Holger Gerhardt, Rachel Golub, Christoph Scheiermann and Julien Y. Bertrand

Dear Julien,

I have now received all the referees' reports on the above manuscript, and have reached a decision. The referees' comments are appended below, or you can access them online: please go to:

As you will see, the referees express considerable interest in your work, but have some significant criticisms and suggestions for improvements to your manuscript. If you are able to revise the manuscript along the lines suggested, I will be happy receive a revised version of the manuscript. Please also note that Development will normally permit only one round of major revision. If it would be helpful, you are welcome to contact us to discuss your revision in greater detail. Please send us a point-by-point response indicating your plans for addressing the referees' comments, and we will look over this and provide further guidance.

Please attend to all of the reviewers' comments and ensure that you clearly highlight all changes made in the revised manuscript. Please avoid using 'Tracked changes' in Word files as these are lost in PDF conversion. I should be grateful if you would also provide a point-by-point response detailing how you have dealt with the points raised by the reviewers in the 'Response to Reviewers' box. If you do not agree with any of their criticisms or suggestions please explain clearly why this is so.

Reviewer 1

Advance summary and potential significance to field

The manuscript entitled "BMAL1 modulates glutamine supply to control hematopoietic stem and progenitor cell expansion" describes a new role for the circadian clock TF BMAL1 in hematopoietic stem and progenitor cell development during zebrafish embryogenesis. Using genetic and chemical modulation in zebrafish, the authors claim that BMAL1 regulates HSPC expansion but not formation during development. Mechanistically, they make connections between BMAL1 to proliferation and glutamine metabolism although the data supporting these connections are somewhat thin. Additionally, there are some discrepancies in the text with regard to when BMAL1 is deployed during HSPC development. Moreover, they include data from murine fetal liver to test conservation, yet most of the data disagree with the zebrafish data. Additional experiments and changes to some of the claims in the results and discussion are needed to further support the conclusions.

Comments for the author

- 1) There is inconsistent language used to describe which timepoints in zebrafish development correlate to HSPC expansion, migration, and proliferation. For example, colonization of HSPC is still ongoing at 60 hpf
- 2) The authors claim the DNBMAL1-driven difference in HSPC levels within the CHT at 60 hpf is due to proliferation yet the quantification of phosphoH3 is performed at 72 hpf. The changes detected were extremely subtle. The data support a potential dual role for BMAL1. The early phenotype detected by 60 hpf could be acting to promote migration of nascent HSPC from the DA to the CHT, which is partially supported by the early trend of increased *cmyb*⁺ cells in the CHT as early as 54 hpf. The second effect could be on HSPC proliferation and/or retention of cells once in the CHT. Proliferation rates at later time points should be performed to test this model. Additionally, the accumulation of HSPC at 4.5 and 7 dpf within the DNBMAL1 overexpressing CHT could be caused by retaining the cells here. The result would be fewer HSPC within the thymus and kidney at 6-7dpf. These alternative possibilities need to be explored.
- 3) The connection of BMAL1 to glutamine metabolism as a means to regulate HSPC levels during development is an interesting finding but not strongly supported. Although treatment with L-leucine had similar effects on zebrafish HSPC levels and murine fetal liver cells, the connection to BMAL1 does not seem conserved. The authors suggest it is BMAL1 regulation of *glud1a* in zebrafish that increases glutamine levels and drives HSPC expansion, yet murine *Glud1a* is not altered in murine BMAL1 KO mice. Moreover the experiments testing the effects of glutamine modulation on HSPC levels are only performed in WT animals. Genetic suppression experiments in EC-DNBMAL1 zebrafish where *glud1a* is overexpressed or treated with the *Glud1* activator L-leucine are needed to strengthen the proposed mechanism.
- 4) In the murine FL culture experiments, there is a possibility that the cells treated with L-leucine could be dying more in the absence of glutamine. It is important to provide viability assessments for these experiments.
- 5) The introduction is very focused on BMAL1 in terms of the circadian clock, however there is not much about glutamine/glutamate. As the results barely focus on the circadian clock it would be beneficial to include more introduction on topics covered more in the results.

Reviewer 2

Advance summary and potential significance to field

The work by Tim Petzold and co-authors builds up on previous reports from Julien Bertrand's laboratory aimed at understanding metabolic aspects of the niche/hematopoietic stem and progenitor cells (HSPCs) interplay, in devoted niches during development (ex: Mahony et al 2016; Xue et al 2017, see the manuscript Ref list). This is an important topic, of significant relevance for understanding fundamental aspects of HSPC development and stemness maintenance, as well as pathological aspects leading to abnormal growth and increased propensity to mutations leading to malignancies. More specifically, the current paper investigates the potential function of *bmal1*, an essential component of the circadian clock transcriptional machinery, in the control of HSPC expansion by endothelial cells of the caudal hematopoietic tissue (CHT), in the zebrafish embryo/larva. For their functional studies, they have developed UAS transgenic lines that express either dominant negative or point mutation mutant forms of the *bmal1a* isoform (DN-*bmal1a*). Upon

expression of the DN form in CHT endothelial cells, they report on a phenotype on HSPC proliferation, in the early larva. This phenotype is further analyzed using bulk RNAseq on sorted cells, with evidence for the *bmal1a*-mediated regulation of genes involved in glutamine homeostasis. Finally, they complement their work by addressing this metabolic pathway, via genetic interference, in the mouse, and show that hepatocytes rather than endothelial cells exert this function.

While the experimental flow of the paper is well conducted in general, and the work of quite significant interest for the field, there are important points, both on conceptual aspects as well as on potential experimental caveats, that should be considered and revised to strengthen the experimental outcome and the conclusions of the study.

Comments for the author

Major points

1. Regarding the function of the veinous tissue as a niche for HSPCs, the time windows of endothelial expression of *bmal1a* and *bmal1b* (the 2 isoforms expressed in the zebrafish) is significantly ahead of the time window of HSPC emergence (HSPC being produced during the definitive wave of hematopoiesis leading to the production of HSCs that spans over approximately 28 to 72 hpf embryo-early larva developmental period); in Fig. 1a, *bmal1a* is not detected anymore at 48hpf which is the peak of definitive emergence (Kissa and Herbomel 2010), which appears to be even shorter for *bmal1b* (see Fig. S2A). In addition, the apparent absence of effect of DN-*bmal1a* on HSPC cell number using the marker *cmyb* in in situ hybridization at 48, 54 hpf (Fig. S5C, D), but apparent effect on cell number at 72 hpf (Fig. 2C using *cmyb*:eGFP cell numbering; Fig. S6A, B; using *cd41*:eGFP and *runx1*:nls-mCherry cell numbering), and of *cmyb* signal at 4.5 and 7 dpf (respectively Fig. 2B and Fig. S6C, using in situ *cmyb* signal) is even less fitting the *bmal1* isoforms expression windows. So is the case also for *glud1a*, one of the target of *bmal1a* (Fig. 4E, not detected anymore at 60 hpf) and for the transporters illustrated Fig. S10C that are not anymore detected (in situs only are shown and not qPCR) at 72 hpf when it is at that time point the authors have performed their proliferation assay Fig. 3; why is it so?

One obvious explanation about asynchronies may be that the data illustrate mRNA expression and not protein, although 4.5/7 dpf is a long gap after 30 hpf and one would have to assume that the half-life of the protein is rather long. Could the authors comment on that? Is there any antibody available to test the *bmal1a*, *b* protein expression?

In addition, in relation to the apparent lack of rhythmicity of expression of *bmal1a* in the CHT raised in light-dark cycles (Fig. 1B) in comparison to the head (Fig. S1), could it be that the regulation takes place at the protein rather than at the mRNA level? Do the authors know if *bmal1a* and *b* can be ubiquitinated (or any other modification) for degradation?

Finally, the authors could try to provide a clue on the regulation of *bmal1a* and/or *b* stability at the protein level by expressing a heat-shock (*hsp70*) inducible *bmal1a* and/or *b* (both for the full length and for their DN forms) fused to a small epitope tag to detect the expression by immunofluorescence.

2. The authors conclude that *bmal1a*, *b* are not expressed with rhythmicity in the tail and that the function of *bmal1a* in the regulation of CHT-resident HSPCs would occur in a non-rhythmic manner (as also exemplified in the rat fetal liver, see the Discussion), arguing for a conserved specificity across species. However, embryos/larvae are grown artificially in fish facilities, with light/dark cycles and light intensities that may not fit with the natural environment. What would happen, to investigate extreme situations, if embryos are left either in 100% light or in 100% dark for 3 consecutive days; how would that impact *bmal1a* and *b* mRNA expression? Beside being of fundamental interest, this point may be important for people in the field because in some fish facilities it may happen that embryos/larvae are kept in the dark, for example because of equipment restrictions.

3. It is somehow striking that, in 4.5 dpf larvae, expression of DN-*bmal1a* does not alter neutrophil, erythrocyte, macrophage or T-cell numbers (Fig. S7). How can that be explained if *bmal1a* is involved in regulating HSPC number/proliferation? HSPC being upstream of these cell types (except for macrophages and erythrocytes that are also produced earlier from progenitors, during the non-

definitive wave and during the phase of production of the so-called Erythro-Myeloid Progenitors (EMPs, see Bertrand et al 2007)), one would expect that if they are amplified in the context of the DN expression, this would have an incidence on the number of differentiated downstream progenitors and differentiated cells, isn't it ?

In addition, *mpx*, *gata1*, *mfap4*, *rag1* are expressed by more differentiated progenitors and differentiated cells, with all these cells being dispersed all over the larval body, hence it is inaccurate to focus on the tail only to conclude that these specific cells are not altered (the only exception is for *rag1* since cells from the lymphoid lineage and expressing this marker essentially reside in the thymus at that stage). How many of these cells reside in the AGM for example? In the head for macrophages and neutrophils?

At the same age of 4.5 dpf and upon expression of DN-*bmal1a*, *cmyb* appears to be increased quite significantly in the CHT (ex: Fig. 2B), which differs from the markers above, which is somehow expected for cells that would be higher in the hematopoietic hierarchy (in particular progenitors) and if indeed *bmal1a* controls an axis important for the regulation of HSPCs. However, the impact of this work would be much lowered if this axis concerns (a) more restricted lineage(s). It appears that this may indeed be the case, and that the population that may be more concerned by the *bmal1a*-*glud1a*-glutamine pathway are EMPs or more restricted progenitors of the erythroid lineage. Several arguments are in favor of this: (1) the time-window of *bmal1a* expression (EMP have been proposed to emerge before the wave producing HSCs), (2) the bulk RNAseq data of this paper (performed at 36 hpf, see Fig. 4 and Table S6); comparing the differentially expressed genes in Table S6, many apparently upregulated genes are specific for cells of the erythroid lineage and involved in hemoglobin/heme function (ex: *hbbe*, *hbae* isoforms), attesting for the expansion of cells of this lineage upon overexpression of DN-*bmal1a*. Hence, DN-*bmal1a* may be more restrictively involved in the functions of a niche specific for expansion of erythroid progenitors; at 36 hpf, some erythroid progenitors whose ancestors have emerged from the vascular system slightly earlier are inevitably found in the bulk of *kdrl:Gal4+;UAS:DN-bmal1a+;kdrl:eGFP+* cells sorted by FACS (Fig. 4A). Can the authors think of this possibility and, if pertinent, modify their manuscript accordingly?

4. The experiments performed Fig. 2D are not convincing. This is because of the silencing of the UAS promoter, which is particularly well known in the field for this *gata2b* fish line (unless a Tg line other than the one described in Butko et al 2015 has been obtained, see the Refs list of the manuscript). Hence, because of silencing and also because, with their DN-*bmal1a* line, the authors cannot make sur, for each larva analyzed, that the DN protein is indeed expressed by a significant number of cells, a negative result cannot be interpreted accurately. To be able to check efficiency of the DN protein expression, the authors should have produced a Tg line expressing a DN with an epitope tag, has mentioned point 1 above. Presently and based on these experiments, the authors cannot firmly conclude that *bmal1a* does not have any function intrinsically in hematopoietic cells (and in situ hybridizations do not have the resolution to clearly show that only endothelial cells of the CHT express *bmal1a* and/or *bmal1b*).

5. Several quantitative approaches may not be accurate all along the manuscript.

This concerns:

- In situ hybridizations. Chromogenic signal do not develop linearly, particularly when comparing relatively low and higher signals that may reach saturation (saturation of the enzymatic reaction, saturation of pixels; ex: Fig. 1B, Fig. 2B, Fig S1, Fig. S4E, Fig. S6C, Fig. S7). In addition, how can cell number be accurately measured when the density of signal is extreme (ex: Fig. S5D, Fig. S7C)? The authors should strongly consider combining more and, whenever possible, in situ hybridization images (for qualitative information) and qRT-PCR (such as for example the data presented Figure S9).

- Fluorescence images. We are lacking information on how the image were acquired and treated for quantitative analysis. We do not know if we see stacks or z-projections (not detailed, neither in Figure legends, nor in Methods). Quantitative analyses cannot be accurate on z-projections and should be performed after 3-D reconstitutions from z-stacks.

Fig. S6B using the *runx1:nls-mCherry* fish line exhibits a pronounced background (including pigment autofluorescence most probably), casting doubts on accuracy of quantifications.

Minor points

1. The abstracts is somehow convoluted and would gain being simplified (ex: conclusions on new discovery written rather at the end); anyhow, it should be reduced to 180 words, according to the journal policies.
2. All over the manuscript, the authors do not discriminate between an embryo (before 72 hpf) and a larva (starting at 72 hpf/3 dpf). They should correct this oversimplification.
3. Supplementary Fig. 4, panel C, *dll4* in situ; why these images do not match the ones shown at the same stage Supplementary Fig. 3D?
4. The authors should explain in the text why they stopped using the R88A-DN mutant in their study.
5. I am wondering about the accuracy of the quantifications in experiments Supplementary Figure 3F; this is because eGFP under the control of *kdrl* should also be expressed in hematopoietic cells (we are at the peak of the emergence); how do the authors can than discriminate between endothelial and hematopoietic cells? to some extent, this is also true for panel G although with a lesser impact on accuracy because 4.5 dpf is shifted in time and the number of emergences has dropped dramatically.
6. Supplementary Figure 3A and its legend + related Methods. I should guess the authors did not clone the DNA with introns but that, to build their constructs (full length and deletion mutant), they obtained a cDNA, isn't it? The legend should be clearer, and the method section substantiated. On this line, the authors should put an effort substantiating all their Figure legends that are too short. Writing all what we can find in the panels is not sufficient for the reader to precisely capture the conditions of the experiments.

Reviewer 3

Advance summary and potential significance to field

The role of circadian clock in HSPC expansion remains unclear. The authors used zebrafish and mouse models to tackle this question in this work. They showed that EC-specific dominant negative form of BMAL1 in zebrafish induced increased HSPC expansion in the CHT, but not affecting HSC emergence in the AGM and vessel plexus in the CHT. They further showed that the BMAL1-GLUD1-Glutamine axis is responsible for HSPC expansion phenotype. Interestingly, they also observed this axis also exists in mouse fetal liver.

Overall, I felt this work is interesting and it provides new insights into our understanding of developmental hematopoiesis in fish and mice. However, there are still some issues which should be addressed before further consideration of this work for publication in DEVELOPMENT.

Major concerns:

- 1, since the expression of *bmal1a* is not rhythmic at all in the CHT, is it possible that it plays a non-canonical role in HSPC expansion as a traditional circadian clock gene? Furthermore, any difference in HSPC expansion during light-dark cycles?
- 2, it seems contradictory that *bmal1a* expression only exists between 24-42 hpf but not thereafter, however, the phenotype occurs at 60hpf onwards at the time it is not expressed. Considering its rhythmic expression in the head region as shown in Figure S1, is it possible that *bmal1a* exerts its cell-autonomous role in the central nerve system (CNS) initially, then transduce this long-range signal to the CHT region? This hypothesis is highly reasonable and the crosstalk between neural system and hematopoiesis is well established in the field.
- 3, the authors claimed that although HSPC expansion is enhanced, but the differentiated lineages are not altered. How would this be possible, in other words, where do these increased HSPCs go at later stages?

Minor concerns:

- 1, the *per2* expression seems only decreased a little bit in Supp. Fig 3C, suggesting that the dominant-negative effects of this line not sufficient.

First revision

Author response to reviewers' comments

Comments from the Reviewers:

We thank all the reviewers for their time, as well as their comments and suggestions. We have addressed all the points, as indicated below

Reviewer 1: SUMMARY OF THE ADVANCE MADE IN THIS PAPER AND ITS POTENTIAL SIGNIFICANCE TO THE FIELD

The manuscript entitled "BMAL1 modulates glutamine supply to control hematopoietic stem and progenitor cell expansion" describes a new role for the circadian clock TF BMAL1 in hematopoietic stem and progenitor cell development during zebrafish embryogenesis. Using genetic and chemical modulation in zebrafish, the authors claim that BMAL1 regulates HSPC expansion but not formation during development. Mechanistically, they make connections between BMAL1 to proliferation and glutamine metabolism although the data supporting these connections are somewhat thin. Additionally, there are some discrepancies in the text with regard to when BMAL1 is deployed during HSPC development. Moreover, they include data from murine fetal liver to test conservation, yet most of the data disagree with the zebrafish data. Additional experiments and changes to some of the claims in the results and discussion are needed to further support the conclusions.

SUGGESTIONS TO AUTHORS

1) There is inconsistent language used to describe which timepoints in zebrafish development correlate to HSPC expansion, migration, and proliferation. For example, colonization of HSPC is still ongoing at 60 hpf.

We thank the reviewer for the suggestion. We have changed words / sentences in the text in order to accommodate this suggestion and improve consistency. Changes are highlighted in the manuscript.

2) The authors claim the DNBMAL1-driven difference in HSPC levels within the CHT at 60 hpf is due to proliferation yet the quantification of phosphoH3 is performed at 72 hpf. The changes detected were extremely subtle. The data support a potential dual role for BMAL1. The early phenotype detected by 60 hpf could be acting to promote migration of nascent HSPC from the DA to the CHT, which is partially supported by the early trend of increased *cmyb*⁺ cells in the CHT as early as 54 hpf. The second effect could be on HSPC proliferation and/or retention of cells once in the CHT. Proliferation rates at later time points should be performed to test this model. Additionally, the accumulation of HSPC at 4.5 and 7 dpf within the DNBMAL1 overexpressing CHT could be caused by retaining the cells here. The result would be fewer HSPC within the thymus and kidney at 6-7dpf. These alternative possibilities need to be explored.

We thank the reviewer for highlighting these possibilities; we have investigated the points raised by performing a number of additional experiments.

First, the reviewer's point regarding the wording in the manuscript about proliferating HSPCs at timepoints earlier than the 72 hpf one originally investigated is well-taken. To address the concern, we performed an additional experiment to evaluate whether HSPCs in the CHT proliferate more in *kdrl:GAL4;UAS:DN-bmal1a* embryos relative to controls at 60 hpf. We generated *kdrl:GAL4;UAS:DN-bmal1a;cmyb:GFP* embryos and controls. Then, as was done before at 72 hpf, we performed an immunohistochemistry experiment using anti-GFP and anti-PH3 antibodies before imaging the CHT area of embryos using spinning disk confocal microscopy. Our data show that even at the 60 hpf timepoint, there is a significant increase in the number of proliferating HSPCs in the CHT region imaged in *kdrl:GAL4;UAS:DN-bmal1a;cmyb:GFP* embryos relative to controls. We have included this data in Figure 3A of the new version of the manuscript.

We agree with this reviewer that the data in our previous version of the manuscript do not exclude the possibility that *bmal1a* may also regulate the migration of HSPCs into or out of the CHT in

zebrafish. While we believe that the HSPC proliferation experiment we have now performed at 60 hpf provides strong evidence for an increased HSPC expansion rate in *kdrl:GAL4;UAS:DN-bmal1a* embryos, the possibility remains for an increased migration rate of HSPCs into the niche from the dorsal aorta. As such, we have included this possibility in the discussion as follows, “Whether Bmal1a also plays a role in the regulation of HSPC migration into the CHT in zebrafish will be interesting to investigate in future studies.”

In order to explore whether Bmal1a plays a role in the migration of HSPCs away from the CHT niche, we performed *cmyb in situ* hybridization experiments in *kdrl:GAL4;UAS:DN-bmal1a* larvae at 4.5 and 7 dpf. We then imaged the larvae dorsally in order to analyze *cmyb* expression level and hence HSPC numbers in the kidney glomeruli. We found that at 4.5 dpf there is indeed a decrease in *cmyb* signal in the kidney glomeruli (Supplementary Figure 8A). This difference is absent at 7 dpf (Supplementary Figure 8B). This indicates that in the absence of functional endothelial Bmal1a there is indeed a migration delay in HSPCs away from the CHT to the kidney glomeruli, suggesting that functional Bmal1a (in the CHT vascular niche) actively promotes the migration of HSPCs away from the CHT. We have included this data in the new version of the manuscript and have also included the following line in the discussion “Our data also suggest that Bmal1a promotes the migration of HSPCs away from the CHT.”

3) The connection of BMAL1 to glutamine metabolism to regulate HSPC levels during development is an interesting finding but not strongly supported. Although treatment with L-leucine had similar effects on zebrafish HSPC levels and murine fetal liver cells, the connection to BMAL1 does not seem conserved. The authors suggest it is BMAL1 regulation of *glud1a* in zebrafish that increases glutamine levels and drives HSPC expansion, yet murine *Glud1a* is not altered in murine BMAL1 KO mice. Moreover the experiments testing the effects of glutamine modulation on HSPC levels are only performed in WT animals. Genetic suppression experiments in EC-DNBMAL1 zebrafish where *glud1a* is overexpressed or treated with the *Glud1* activator L-leucine are needed to strengthen the proposed mechanism.

Regarding conservation of the molecular mechanism proposed in our manuscript, we have suggested in our work that the mechanism at play is indeed not conserved in endothelial cells of the mouse but, as discussed, we suggest conservation of the molecular pathway is present in hepatocytes in mammals. This is supported by the expression of *glud1a* in hepatocytes but not in endothelial cells of human fetal liver, as shown in Supplementary Figure 15 and explains the lack of difference in the expression of *Glud1a* in mouse FL EC BMAL1 KO animals. In previous work from our lab, we encountered similar conserved factors that perform the same functions in the HSPC niche in different species, but are present in different cell types, for example, *IFI30*, expressed by CHT-ECs in zebrafish but found in macrophages neighboring HSPCs in the human fetal liver (Cacialli, Nat Comms 2021).

In order to strengthen the proposed mechanism, we treated *kdrl:GAL4;UAS:DN-bmal1a* embryos and controls with L-leucine, BCH or ADP between 48 hpf and 4.5 dpf (following HSPC specification) before carrying out *cmyb in situ* hybridization and quantifying the signal area as was done previously throughout the manuscript. While L-leucine resulted in a non-significant reduction ($p=0.08$) in *cmyb* signal compared with controls (shown below), BCH and ADP treatment resulted in a significant reduction in *cmyb* signal area and hence HSPC numbers in the CHT (Figure 5B and C). This makes sense mechanistically, since these compounds are activators of *Glud1a* and hence result in more alpha-ketoglutarate at the expense of glutamine. However, the fact that *glud1a* expression is reduced in *kdrl:GAL4;UAS:DN-bmal1a* animals may also provide an explanation as to why the difference in HSPC numbers in treated vs control *kdrl:GAL4;UAS:DN-bmal1a* larvae is not as dramatic as in WT embryos (compare Figure 5 B and C with Figure 5 E, F and G). This may also be the reason why treating *kdrl:GAL4;UAS:DN-bmal1a* larvae with some compounds (L-leucine) did not result in a significant reduction in HSPC numbers in the CHT (shown below). We thank the reviewer for the suggestions, which we feel has strengthened the proposed molecular mechanism.

4) In the murine FL culture experiments, there is a possibility that the cells treated with L-leucine could be dying more in the absence of glutamine. It is important to provide viability assessments for these experiments.

We thank this reviewer for their suggestion. Cell viability data for the FL culture experiments had been provided (Fig. 6B), which show that there is no difference in cell death when L-leucine or glutamine are added to the culture media in comparison with non-supplemented controls. We apologize that this had not been clear.

5) The introduction is very focused on BMAL1 in terms of the circadian clock, however there is not much about glutamine/glutamate. As the results barely focus on the circadian clock it would be beneficial to include more introduction on topics covered more in the results.

We have added an additional sentence in the introduction in order to provide more detail on the association between the circadian clock and the expression of genes involved in glutamine metabolism. We also added a new paragraph on the known roles of glutamine during myelopoiesis and erythropoiesis. We now think that the introduction flows better and includes sufficient detail on all topics covered in the manuscript in order to set the scene for the remainder of our manuscript

Reviewer 2: SUMMARY OF THE ADVANCE MADE IN THIS PAPER AND ITS POTENTIAL SIGNIFICANCE TO THE FIELD

The work by Tim Petzold and co-authors builds up on previous reports from Julien Bertrand's laboratory aimed at understanding metabolic aspects of the niche/hematopoietic stem and progenitor cells (HSPCs) interplay, in devoted niches during development (ex: Mahony et al 2016; Xue et al 2017, see the manuscript Ref list). This is an important topic, of significant relevance for understanding fundamental aspects of HSPC development and stemness maintenance, as well as pathological aspects leading to abnormal growth and increased propensity to mutations leading to malignancies. More specifically, the current paper investigates the potential function of *bmal1*, an essential component of the circadian clock transcriptional machinery, in the control of HSPC expansion by endothelial cells of the caudal hematopoietic tissue (CHT), in the zebrafish embryo/larva. For their functional studies, they have developed UAS transgenic lines that express either dominant negative or point mutation mutant forms of the *bmal1a* isoform (DN-*bmal1a*). Upon expression of the DN form in CHT endothelial cells, they report on a phenotype on HSPC proliferation, in the early larva. This phenotype is further analyzed using bulk RNAseq on sorted cells, with evidence for the *bmal1a*-mediated regulation of genes involved in glutamine homeostasis. Finally, they complement their work by addressing this metabolic pathway, via genetic interference, in the mouse, and show that hepatocytes rather than endothelial cells exert this function.

While the experimental flow of the paper is well conducted in general, and the work of quite significant interest for the field, there are important points, both on conceptual aspects as well as on potential experimental caveats, that should be considered and revised to strengthen the experimental outcome and the conclusions of the study.

SUGGESTIONS TO AUTHORS

Major points

1. Regarding the function of the venous tissue as a niche for HSPCs, the time windows of endothelial expression of *bmal1a* and *bmal1b* (the 2 isoforms expressed in the zebrafish) is significantly ahead of the time window of HSPC emergence (HSPC being produced during the definitive wave of hematopoiesis leading to the production of HSCs that spans over approximately 28 to 72 hpf embryo- early larva developmental period); in Fig. 1a, *bmal1a* is not detected anymore at 48hpf which is the peak of definitive emergence (Kissa and Herbomel 2010), which appears to be even shorter for *bmal1b* (see Fig. S2A). In addition, the apparent absence of effect of DN-*bmal1a* on HSPC cell number using the marker *cmyb* in in situ hybridization at 48, 54 hpf (Fig. S5C, D), but apparent effect on cell number at 72 hpf (Fig. 2C using *cmyb*:eGFP cell numbering; Fig. S6A, B; using *cd41*:eGFP and *runx1*:nls- mCherry cell numbering), and of *cmyb* signal at 4.5 and 7 dpf (respectively Fig. 2B and Fig. S6C, using in situ *cmyb* signal) is even less fitting the *bmal1* isoforms expression windows. So is the case also for *glud1a*, one of the target of *bmal1a* (Fig. 4E, not detected anymore at 60 hpf) and for the transporters illustrated Fig. S10C that are not anymore detected (in situs only are shown and not qPCR) at 72 hpf when it is at that time point the authors have performed their proliferation assay Fig. 3; why is it so?

One obvious explanation about asynchronies may be that the data illustrate mRNA expression and not protein, although 4.5/7 dpf is a long gap after 30 hpf and one would have to assume that the half- life of the protein is rather long. Could the authors comment on that? Is there any antibody available to test the *bmal1a*, *b* protein expression?

We thank the reviewer for the points raised. With regards to the expression of the glutamine transporters, we found that overall, the expression of the genes encoding these is quite low (as is often the case for other transporter genes) which is why we performed qPCR for these along with the *in situ* hybridizations (which is not as sensitive). The qPCR analysis for glutamine transporter gene expression in endothelial cells and HSPCs in the tail was performed at 48 hpf. We included the *in situ* hybridization data for a subset of the transporter genes which were found to be expressed in the qPCR since we wanted to demonstrate their expression can also, at least to some extent, be demonstrated in the CHT using this technique. Due to the low sensitivity rate of *in situ* hybridizations however, our data does not exclude that these transporter genes may also be expressed at 72 hpf (and even later) in both endothelial cells and HSPCs.

Regarding the timings of *bmal1a* and *bmal1b* expression, our data demonstrate that it is in fact this loss of *bmal1a* expression (as well as *bmal1b*) after 42 hpf that permits the HSPCs to greatly expand in number in the CHT. Mechanistically, we have shown that the loss of *bmal1a* expression results in a downregulation of *glud1a* expression (which, as the reviewer correctly points out is then also lost). This, in turn, results in an increase in the concentration of glutamine in the CHT (HSPC niche), as demonstrated and discussed in our manuscript, since glutamate is converted to glutamine at the expense of alpha-ketoglutarate. This increase in CHT glutamine concentration fuels HSPC expansion in the CHT. HSPCs begin to arrive in the CHT from around 48 hpf onwards. The lag period between 48 hpf and the phenotype being observed from 60 hpf onwards can then be explained further as it will take time for the newly arrived HSPCs to then take up glutamine, for the relevant signaling pathways to be induced and for cell proliferation to take place.

Concerning protein detection, we have been unable to find any reliable tools to assess *bmal1a/b* protein content since, to the best of our knowledge, there is no commercial antibody available for fish. We only found one antibody, made by Chao Liu et al, Sci Reports 2015, which was only used for immunoprecipitation assays.

In addition, in relation to the apparent lack of rhythmicity of expression of *bmal1a* in the CHT raised in light-dark cycles (Fig. 1B) in comparison to the head (Fig. S1), could it be that the regulation takes place at the protein rather than at the mRNA level? Do the authors know if *bmal1a* and *b* can be ubiquitinated (or any other modification) for degradation?

This is an interesting suggestion. There are indeed reports of BMAL1 ubiquitination, as well as other modifications of the BMAL1 protein. Hence, BMAL1 may be regulated differently on a post-

translational level in different tissues. However, we have not investigated these aspects of the biology of BMAL1 in our manuscript, and our results clearly demonstrate that there is non-rhythmic expression of *bmal1a* in the CHT of embryos (Figure 1A) while expression of *bmal1a* oscillates in the heads of these embryos (Supplementary Figure 1).

Finally, the authors could try to provide a clue on the regulation of *bmal1a* and/or *b* stability at the protein level by expressing a heat-shock (*hsp70*) inducible *bmal1a* and/or *b* (both for the full length and for their DN forms) fused to a small epitope tag to detect the expression by immunofluorescence.

Although this is an interesting proposition, understanding how exactly BMAL1 is regulated molecularly on the protein level has been the topic of investigation in many other studies, but it is not the focus of our present manuscript.

2. The authors conclude that *bmal1a*, *b* are not expressed with rhythmicity in the tail and that the function of *bmal1a* in the regulation of CHT-resident HSPCs would occur in a non-rhythmic manner (as also exemplified in the rat fetal liver, see the Discussion), arguing for a conserved specificity across species. However, embryos/larvae are grown artificially in fish facilities, with light/dark cycles and light intensities that may not fit with the natural environment. What would happen, to investigate extreme situations, if embryos are left either in 100% light or in 100% dark for 3 consecutive days; how would that impact *bmal1a* and *b* mRNA expression? Beside being of fundamental interest, this point may be important for people in the field because in some fish facilities it may happen that embryos/larvae are kept in the dark, for example because of equipment restrictions.

These aspects have been previously investigated in a number of studies (including Chen et al, Cell Reports, 2023; Yang et al, Cellular and Molecular Life Sciences, 2025; Jensen et al, Cell Reports, 2012). These publications include data for the expression of *bmal1a* and *bmal1b* but also for other circadian rhythm genes when embryos are raised in light-dark, light-light or dark-dark cycles (see Yang et al, figure 3 and supplementary figures 2, 3 and 4; Jensen et al figures 3I). The conclusions of the data in these publications are that if wild-type embryos are raised in light-dark, light-light, or dark-dark conditions, oscillations of circadian rhythm genes, including the clock genes *bmal1a* and *bmal1b* sometimes remain, although in some cases at a reduced amplitude. Some of these reports in which zebrafish embryos were raised in light-light or dark-dark conditions seem to result in the loss of the rhythmic expression of circadian genes (Chen et al, Cell Reports, 2023). Together, this points to a complex picture in which the precise timings and intensity of light in which embryos are raised likely results in differing outcomes when it comes to the expression of circadian rhythm genes in zebrafish larvae, at least in embryos / larvae globally. In our manuscript, however, we established an experimental setup which induced rhythmic *bmal1a* gene expression in the heads of embryos (supplementary figure 1) while the expression of *bmal1a* in the tails of the same embryos is not rhythmic.

While we have not investigated specifically what happens to the expression of *bmal1a* and *bmal1b* if embryos are left in either 100% light or 100% dark conditions, we would not expect their expression to differ in the CHT in relation to embryos raised in 12hr light:12hr dark cycles, since the presence/absence of light when raised in 12hr light:12hr dark did not alter the direction in expression. However, we investigated the downstream effects of extreme light conditions on HSPC numbers by raising embryos in either 12hr light:12hr dark cycles, in constant darkness or in constant light. We then fixed these embryos at 72 hpf and carried out WISH for *cmlyb* to determine HSPC numbers. We quantified *cmlyb* WISH signal in the CHT region, as was carried out in the remainder of our manuscript. We found that there was no difference in the number of HSPCs when embryos are raised in constant darkness relative to embryos raised in 12hr light:12hr dark cycles. However, intriguingly, we detected an increase in *cmlyb* signal in the CHT (and hence an increase in HSPC numbers) when embryos were raised in constant light in comparison to those raised in 12hr light:12hr dark cycles (see data below). Since *bmal1a* expression (and the expression of other circadian clock genes investigated) is not rhythmic in the CHT when embryos are raised in 12hr light:12hr dark cycles, we do not believe this phenotype is related to an expansion difference caused by *bmal1a* in the CHT. However, we have a number of other hypotheses as to what may be the driver of this phenotype; 1. It may be that the developmental rate of the embryo is increased when raised in constant light. 2. There may be an increase in the rate/amount of HSPC

specification when embryos are raised in constant light. 3. There may be a yet unknown light-responsive factor present in the CHT which also plays a role in controlling HSPC expansion. While investigating these hypotheses is beyond the scope of the current manuscript, it will be interesting to investigate them further in future work.

3. It is somehow striking that, in 4.5 dpf larvae, expression of DN-*bmal1a* does not alter neutrophil, erythrocyte, macrophage or T-cell numbers (Fig. S7). How can that be explained if *bmal1a* is involved in regulating HSPC number/proliferation? HSPC being upstream of these cell types (except for macrophages and erythrocytes that are also produced earlier from progenitors, during the non-definitive wave and during the phase of production of the so-called Erythro-Myeloid Progenitors (EMPs, see Bertrand et al 2007)), one would expect that if they are amplified in the context of the DN expression, this would have an incidence on the number of differentiated downstream progenitors and differentiated cells, isn't it ?

We thank the reviewer for asking this important question. We agree that it is an intriguing finding that there are no differences in any of the differentiated cell numbers in EC-specific DN-*Bmal1a* zebrafish larvae. We believe that the most likely explanation for this is that, as has been recently reported, HSPCs contribute minimally to the pool of differentiated cells at these early developmental stages (Ulloa et al, Cell Reports, 2021 and Ferrero et al, Development 2021). We have added this hypothesis in the manuscript text in the relevant section. A number of other hypotheses also exist. For example, it may be that the expanded HSPCs differentiate later (differentiated cell numbers were quantified at 4.5 dpf) or it could also be that they don't have the same differentiation potential in this particular context. One further hypothesis is that the HSPC expansion rate exceeds the differentiation rate. It will definitely be interesting to investigate in future studies.

In addition, *mpx*, *gata1*, *mfap4*, *rag1* are expressed by more differentiated progenitors and differentiated cells, with all these cells being dispersed all over the larval body, hence it is inaccurate to focus on the tail only to conclude that these specific cells are not altered (the only exception is for *rag1* since cells from the lymphoid lineage and expressing this marker essentially reside in the thymus at that stage). How many of these cells reside in the AGM for example? In the head for macrophages and neutrophils?

We thank this reviewer for pointing this out. We agree that we have used a reductionist approach by investigating the CHT region as a proxy for the number of *mpx*, *gata1* and *mfap4* positive cells while using the thymus for *rag1* positive cells. As the reviewer rightly points out, this excludes other regions of the larvae in which these cells are also found. However, many other papers in the field quantify differentiated cells in the CHT region and thymus as a proxy for their total numbers, just as we have done, since these are regions of the larvae for which imaging and quantification are easy to perform. Furthermore, *bmal1a*, *glud1a* and *glula* are all expressed in the CHT region. Since we modified the activity of *Bmal1a* in our manuscript which in turn results in a difference in the expression / activity of *glud1a* and *Glula*, respectively, we also expected that if there were changes in the numbers of differentiated cell types, these would most probably be predominantly local changes, within the CHT microenvironment, which is where they differentiate. Together,

these are the reasons as to why we decided to perform most of the experiments in which we quantified differentiated cells (as well as HSPCs) in the CHT.

At the same age of 4.5 dpf and upon expression of DN-bmal1a, *cmyb* appears to be increased quite significantly in the CHT (ex: Fig. 2B), which differs from the markers above, which is somehow expected for cells that would be higher in the hematopoietic hierarchy (in particular progenitors) and if indeed *bmal1a* controls an axis important for the regulation of HSPCs. However, the impact of this work would be much lowered if this axis concerns (a) more restricted lineage(s). It appears that this may indeed be the case, and that the population that may be more concerned by the *bmal1a*-*glud1a*- glutamine pathway are EMPs or more restricted progenitors of the erythroid lineage. Several arguments are in favor of this: (1) the time-window of *bmal1a* expression (EMP have been proposed to emerge before the wave producing HSCs), (2) the bulk RNAseq data of this paper (performed at 36 hpf, see Fig. 4 and Table S6); comparing the differentially expressed genes in Table S6, many apparently upregulated genes are specific for cells of the erythroid lineage and involved in hemoglobin/heme function (ex: *hbbe*, *hbae* isoforms), attesting for the expansion of cells of this lineage upon overexpression of DN-*bmal1a*. Hence, DN-*bmal1a* may be more restrictively involved in the functions of a niche specific for expansion of erythroid progenitors; at 36 hpf, some erythroid progenitors whose ancestors have emerged from the vascular system slightly earlier are inevitably found in the bulk of *kdrl:Gal4+;UAS:DN-bmal1a+;kdrl:eGFP+* cells sorted by FACS (Fig. 4A). Can the authors think of this possibility and, if pertinent, modify their manuscript accordingly?

This is an interesting proposition. We do not think that *Bmal1a* acts specifically on lineage-restricted (eg. erythroid lineage restricted) cells, since we provide robust evidence throughout our manuscript pointing to a role for *Bmal1a* in regulating HSPCs (including in figures 2B-C, 3A-B, Supplementary figure 4, Supplementary figure 5A-C), which are also evidenced by counting live cells at 72 hpf with the *cmyb:GFP*, *cd41:GFP* and *runx1:mCherry* transgenic lines (Figure 2C and Supp Figure 6A-B). However, we agree that there is the possibility that *Bmal1a* may also, in addition to its effect on HSPCs, regulate other cells in the niche, perhaps including more lineage-restricted progenitors, such as EMPs.

To test this possibility, we assessed the glutamine pathway discovered in our manuscript in more detail, using *mindbomb* (*mib*) mutants and controls. EMPs are specified as normal in *mib* embryos, while HSPC specification does not occur (Bertrand et al, Blood 2010). We treated *mib* mutants and sibling controls with glutamine or L-leucine before fixing at 48 or 72 hpf and performing WISH for *mpx* (a neutrophil marker gene). We then quantified the number of *mpx+* cells in the CHT of *mib* mutants and sibling controls. At 48 hpf, the number of *mpx+* cells in the CHT of *mib* siblings did not change when treated with glutamine or L-leucine (both at 1mM). Similarly, in *mib* mutants, L-leucine had no effect on the number of *mpx+* cells, while glutamine treatment resulted in a small decrease in *mpx+* cells in the CHT (see data below). At 72 hpf, the *mpx in situ* signal area in the CHT and thus neutrophil number in this location only increased when treating sibling embryos with glutamine from 16 hpf, but interestingly, this effect is absent when treating siblings from 24 hpf, suggesting that the phenotype observed when treatment with glutamine at 16 hpf is likely due to an increased primitive but not definitive neutrophil proliferation rate in these embryos, although it could also affect EMPs.

Together, this data shows that in the absence of HSPCs (in *mib* mutants), glutamine levels may play a role in regulating neutrophils derived independently of aorta-derived HSPCs in the CHT (although, curiously, there may actually be a reduction and not an increase in neutrophils in the presence of additional glutamine), whilst the addition of compounds such as L-leucine, which modulates *Glud1a* activity, resulting in less glutamine, does not have an effect on the number of *mpx+* cells that are HSPC-independent. This data supports the claims we make in our manuscript that the molecular pathway we have uncovered in our study regulates HSPCs but leaves open the possibility that other HSPC-independent (more lineage restricted) cell types may also be regulated by this pathway. Currently however, we do not have any strong evidence to support this and have therefore decided to not add this piece of data to the manuscript, but this will be very interesting to investigate further in future work.

4. The experiments performed Fig. 2D are not convincing. This is because of the silencing of the UAS promoter, which is particularly well known in the field for this *gata2b* fish line (unless a Tg line other than the one described in Butko et al 2015 has been obtained, see the Refs list of the manuscript). Hence, because of silencing and also because, with their DN-*bmal1a* line, the authors cannot make sur, for each larva analyzed, that the DN protein is indeed expressed by a significant number of cells, a negative result cannot be interpreted accurately. To be able to check efficiency of the DN protein expression, the authors should have produced a Tg line expressing a DN with an epitope tag, has mentioned point 1 above. Presently and based on these experiments, the authors cannot firmly conclude that *bmal1a* does not have any function intrinsically in hematopoietic cells (and *in situ* hybridizations do not have the resolution to clearly show that only endothelial cells of the CHT express *bmal1a* and/or *bmal1b*).

We thank the reviewer for their concern regarding the silencing of this particular line. We agree that it is important to understand whether this is occurring using the *gata2b* line that we have in our lab. We investigated this in detail by crossing our *gata2b*:KALTA4 line with our UAS:DN-*bmal1a* line and to then perform *in situ* hybridization experiments to quantify *bmal1a* expression in double-positive embryos and controls. By analyzing *bmal1a* expression at 4 dpf, we found that in the double-positive *gata2b*:KALTA4 UAS:DN-*bmal1a* embryos, expression was present in the CHT (where HSPCs are residing) but expression was lacking in this region in control embryos (see data below). This result demonstrates that the *gata2b*:KALTA4 line that we used in our study is not silenced. We hope this data satisfies this reviewer.

bmal1a
5. Several quantitative approaches may not be accurate all along the manuscript. This concerns:

- In situ hybridizations. Chromogenic signal do not develop linearly, particularly when comparing relatively low and higher signals that may reach saturation (saturation of the enzymatic reaction, saturation of pixels; ex: Fig. 1B, Fig. 2B, Fig S1, Fig. S4E, Fig. S6C, Fig. S7).

This is true, while staining always differs between *in situ* hybridization experiments performed, the staining period is always kept the same between different conditions in one particular experiment, allowing comparisons to be made between the conditions. Hence, while it is true that signal intensity may not correlate well with expression of a gene, due to the lack of linear development of chromogenic signal (as stated by the reviewer), we believe it is still valid to use this approach to compare expression levels between different conditions. *in situ* hybridization signal has also been quantified in many prior publications in our field (Gherzi et al, Nature Cell Biology 2023, Soto et al, Stem Cell Reports 2021, Potts et al, Cell Reports 2022 etc) in order to understand gene expression differences between conditions and hence we believe it is also a valid approach to use in our study.

In addition, how can cell number be accurately measured when the density of signal is extreme (ex: Fig. S5D, Fig. S7C)?

We thank the reviewer for this question. In order to quantify cell numbers in the regions of interest of *in situ* hybridization images, we first inverted the images in order to increase contrast, before enlarging the regions of interest and also sharpening the images. This allowed us to accurately count individual cells manually whilst excluding background signal, as was done in previous publication from our group (Cacialli et al, Nature Comms 2021; Cacialli et al, EMBO Journal, 2022).

We apologize that this information was missing in our manuscript and so have added detail to the methods section to better explain this. We have also added the citations above. We thank the reviewer for this question which we believe strengthened the story.

The authors should strongly consider combining more and, whenever possible, in situ hybridization images (for qualitative information) and qRT-PCR (such as for example the data presented Figure S9).

In our study, we decided it was necessary to perform the qRT-PCR for *glud1a* specifically, since this was a key gene involved in the molecular mechanism which we investigated downstream of our RNA-seq experiment. Quantification of *in situ* hybridizations was done when we wanted to also include spatial information, which is unfortunately lost when doing most qRT-PCR experiments.

We decided to perform qRT-PCR only when *in situ* hybridization was not sensitive enough and hence signal did not provide us with robust data. We have used this systematic approach successfully in previous publications (Cacialli et al, Nature Comms 2021; Cacialli et al, EMBO Journal, 2022) and hence we hope this provides a satisfactory explanation as to why we decided to again use this strategy when choosing which of these assays to perform in this study.

Fluorescence images. We are lacking information on how the image were acquired and treated for quantitative analysis. We do not know if we see stacks or z-projections (not detailed, neither in Figure legends, nor in Methods). Quantitative analyses cannot be accurate on z-projections and should be performed after 3-D reconstitutions from z-stacks.

We apologize for the lack of information as we used different approaches. In the revision experiment (data which we used to generate Figure 3A), we generated maximum projections from Z-stacks. For all other fluorescence imaging, we did not perform Z-projections or stacks. For these fluorescence images, we took 2D images and performed quantifications using these, as we have done previously (Mahony et al, Blood Advances 2021; Cacialli et al, Nature Comms 2021; Cacialli et al, EMBO Journal, 2022; Cacialli et al, Stem Cell Reports 2023). We appreciate that these details were not mentioned in our manuscript and so have clarified this in the methods section. We thank the reviewer for pointing this out.

Fig. S6B using the *runx1:nls-mCherry* fish line exhibits a pronounced background (including pigment autofluorescence most probably), casting doubts on accuracy of quantifications.

Similar to above, in order to quantify *runx1:nls-mCherry* in the regions of interest, we first inverted the images in order to increase contrast, before enlarging the regions of interest and also sharpening the images. This allowed us to accurately count individual cells manually while excluding pigments and autofluorescence, as was done in previous publications from our group (Cacialli et al, Nature Comms 2021; Cacialli et al, EMBO Journal, 2022). We appreciate that this information was lacking in our manuscript and so have added this information to the methods section. Furthermore, we have also added the citations above.

Minor points

1. The abstracts is somehow convoluted and would gain being simplified (ex: conclusions on new discovery written rather at the end); anyhow, it should be reduced to 180 words, according to the journal policies.

We have simplified and shortened the abstract as requested by removing the following sentence from the end:

“Our findings advance the understanding of the role of BMAL1 in HSPC development, which may contribute to enhancing current regenerative medicine protocols, by improving human HSPC expansion *ex vivo*.”

The abstract is now 172 words in length. We thank the reviewer for the suggestion.

2. All over the manuscript, the authors do not discriminate between an embryo (before 72 hpf) and a larva (starting at 72 hpf/3 dpf). They should correct this oversimplification.

We thank the reviewer for this suggestion and apologize for this oversight. We have made the relevant changes throughout the manuscript - when the zebrafish used were 72 hpf or older, we have changed the word “embryo” to “larvae” or “zebrafish”.

3. Supplementary Fig. 4, panel C, *dll4* *in situ*; why these images do not match the ones shown at the same stage Supplementary Fig. 3D?

The staining in Supplementary Fig 3D was performed for a longer period than that in Supplementary Fig 4C. Staining always differs between *in situ* experiments performed, but the staining period is always kept the same between different conditions of one particular

experiment. Hence, this means that conditions between experiments can be compared but staining intensity with the same *in situ* hybridization probe between different experiments may look different as is the case for *dll4* in these 2 figures. We hope this provides sufficient explanation.

4. The authors should explain in the text why they stopped using the R88A-DN mutant in their study.

We agree and thank the reviewer to their suggestion. We have addressed this in the text by adding that since we saw the same phenotype in the R88A-DN line as in the DN BMAL1A (-transactivation domain) line we felt it was sufficient to perform all remaining experiments of the study with one of the 2 dominant-negative lines that we generated. Since we had already performed some subsequent experiments with the DN BMAL1A (-transactivation domain) line at the point of making this decision, we decided to use this particular line for all subsequent experiments.

5. I am wondering about the accuracy of the quantifications in experiments Supplementary Figure 3F; this is because eGFP under the control of *kdrl* should also be expressed in hematopoietic cells (we are at the peak of the emergence); how do the authors can than discriminate between endothelial and hematopoietic cells? to some extent, this is also true for panel G although with a lesser impact on accuracy because 4.5 dpf is shifted in time and the number of emergences has dropped dramatically.

This is an excellent point. We agree with the reviewer that, especially at 48 hpf, there may be some *kdrl*:EGFP positive hematopoietic cells in the CHT that may also be counted in the analysis and this is a caveat of this experiment. However, we also agree that at 4.5 dpf, hematopoietic cells will likely have little or no impact on the analysis when counting endothelial cells in the CHT region. Hence, we believe the inclusion of the 4.5 dpf timepoint makes our data sufficiently robust to conclude that endothelial cell number does not differ in the CHT between the genotypes investigated. We have made alterations to the text of the manuscript to highlight the potential caveat of the experiment in question and would like to take the opportunity to thank the reviewer for pointing this out.

6. Supplementary Figure 3A and its legend + related Methods. I should guess the authors did not clone the DNA with introns but that, to build their constructs (full length and deletion mutant), they obtained a cDNA, isn't it?

We thank the reviewer for this comment which is absolutely correct. We apologize for the confusion in the original figure. We have re-made the figure that this reviewer is referring to; this time excluding the introns and have put the new figure in the new manuscript version. Additionally, we have updated the legend of this figure to clarify that the DNA fragment to make the dominant-negative construct came from cDNA.

The legend should be clearer, and the method section substantiated.

We have done our best to simplify the figure legends as much as possible. We have also added detail to the methods section in response to this reviewer's suggestion and also including information that was specifically asked about by all 3 reviewers. We feel that this additional information has greatly enhanced the quality of our manuscript and thank this reviewer for this suggestion.

On this line, the authors should put an effort substantiating all their Figure legends that are too short. Writing all what we can find in the panels is not sufficient for the reader to precisely capture the conditions of the experiments.

We thank this reviewer for their suggestion. We have substantiated the figure legends where appropriate to provide additional information regarding the conditions of the experiments performed.

Reviewer 3: The role of circadian clock in HSPC expansion remains unclear. The authors used

zebrafish and mouse models to tackle this question in this work. They showed that EC-specific dominant negative form of BMAL1 in zebrafish induced increased HSPC expansion in the CHT, but not affecting HSC emergence in the AGM and vessel plexus in the CHT. They further showed that the BMAL1-GLUD1-Glutamine axis is responsible for HSPC expansion phenotype. Interestingly, they also observed this axis also exists in mouse fetal liver.

Overall, I felt this work is interesting and it provides new insights into our understanding of developmental hematopoiesis in fish and mice. However, there are still some issues which should be addressed before further consideration of this work for publication in DEVELOPMENT.

Major concerns:

1. since the expression of *bmal1a* is not rhythmic at all in the CHT, is it possible that it plays a non- canonical role in HSPC expansion as a traditional circadian clock gene?

Yes, we agree that it may well play a non-canonical role. In our manuscript however, we use the phrase “non-rhythmic role of *bmal1a*”, rather than non-canonical one, since, in our opinion, this is a more precise term to use in the context of our particular manuscript. Indeed, we show that *clocka* is also expressed in the CHT area, suggesting that in this context, *Bmal1a* and *Clocka* do work together.

Furthermore, any difference in HSPC expansion during light-dark cycles?

We thank this reviewer for this question. We have now investigated this by raising embryos in either 12hr light:12hr dark cycles, in constant darkness or in constant light. We then fixed these embryos at 72 hpf and carried out WISH for *cmyb* to determine HSPC numbers. We quantified *cmyb* WISH signal in the CHT region, as was carried out in the remainder of our manuscript. We found that there is no difference in the number of HSPCs when embryos are raised in constant darkness relative to embryos raised in 12hr light:12hr dark cycles. However, interestingly, we detected an increase in *cmyb* signal in the CHT and hence an increase in HSPC numbers when embryos were raised in constant light in comparison to those raised in 12hr light:12hr dark cycles (see data below). Since *bmal1a* expression (and the expression of other circadian clock genes investigated) is not rhythmic in the CHT when embryos are raised in 12hr light:12hr dark cycles, we do not believe this phenotype is related to an expansion difference caused by *bmal1a* in in the CHT. However, we have a number of other hypotheses as to what may be the driver of this phenotype; 1. It may be that the embryo developmental rate is increased when they are raised in constant light. 2. There may be an increase in the rate/amount of HSPC specification when embryos are raised in constant light. 3. There may be a yet unknown light- responsive factor present in the CHT which also plays a role in controlling HSPC expansion. While investigating these hypotheses is beyond the scope of the current manuscript, we it will be interesting to investigate them further in future work.

2. it seems contradictory that *bmal1a* expression only exists between 24-42 hpf but not thereafter, however, the phenotype occurs at 60hpf onwards at the time it is not expressed.

We thank this reviewer for their question. However, our data demonstrate that, physiologically, this loss of *bmal1a* expression after 42 hpf allows HSPCs to greatly expand in number in the CHT, since the loss of *bmal1a* expression results in a downregulation of *glud1a* expression, which in turn results in an increase in the concentration of glutamine in the CHT. This increase in CHT glutamine levels will then fuel HSPC expansion in the CHT. HSPCs begin to arrive in the CHT from around 48 hpf onwards. The lag period between 48 hpf and the phenotype being observed from 60 hpf onwards could be explained by the time it will presumably take for the newly arrived HSPCs to take up glutamine, for the relevant downstream signaling pathways to be induced and for cell proliferation to take place. Together, the loss of Bmal1a function at earlier stages in *kdrl:GAL4;UAS:DN-bmal1a* animals thus enhances HSPC proliferation in the CHT since a greater concentration of glutamine is already present in the CHT when they arrive in this niche.

Considering its rhythmic expression in the head region as shown in Figure S1, is it possible that *bmal1a* exerts its cell-autonomous role in the central nerve system (CNS) initially, then transduce this long-range signal to the CHT region? This hypothesis is highly reasonable and the crosstalk between neural system and hematopoiesis is well established in the field.

This is an interesting idea, and we agree that there has been a lot of intriguing data recently linking the nervous system with hematopoiesis. However, our data do not provide evidence for this. Our manuscript shows that *bmal1a* is expressed in vascular endothelial cells in the CHT. Even if neurons do express *bmal1a* (our data do not clarify whether this is the case or not, given that Fig S1 only shows *in situ* staining in the heads of embryos as mentioned by this reviewer, but does not discriminate between cell types contributing to *bmal1a* expression in this region), they cannot be the driver of the HSPC expansion phenotype in the CHT of *kdrl:GAL4;UAS:DN-bmal1a* animals, since the DN *bmal1a* construct is expressed only in *kdrl*⁺ cells in the embryos we studied. Furthermore, our data show that downstream transcriptional changes are present in endothelial cells of EC-specific DN BMAL1A embryos. Therefore, while we cannot exclude the possibility of additional mechanisms involving CNS *bmal1a*-mediated control of hematopoiesis in the CHT, our data do not provide any evidence to support this. Although this will be interesting to investigate in future work, we hope that this reviewer can appreciate that this was not the focus of the present study. We are of course however willing to share our tools if other groups would like to test this hypothesis.

3. the authors claimed that although HSPC expansion is enhanced, but the differentiated lineages are not altered. How would this be possible, in other words, where do these increased HSPCs go at later stages?

We thank this reviewer for asking this important question. We agree that it is an intriguing finding that there are no differences in any of the differentiated cell numbers in EC-specific DN-Bmal1a zebrafish larvae. We believe that the most likely explanation is that, as has been recently shown, HSPCs contribute minimally to the pool of differentiated cells at these early developmental stages (Ulloa et al, Cell Reports, 2021 and Ferrero et al, Development 2021), as they rather come from either primitive waves or EMP-derived hematopoiesis. We have added this hypothesis into the manuscript text in the relevant section. Although this is for us the most likely explanation for the lack of difference in differentiated cells, a number of other hypotheses also exist. For example, it may also be that the expanded HSPCs differentiate later (differentiated cell numbers were quantified at 4.5 dpf) or it could also be that they do not have the same differentiation potential in this particular context. One further hypothesis is that the HSPC expansion rate exceeds the differentiation rate. We thank this reviewer again for their important questions regarding this and it will definitely be interesting to investigate these hypotheses in future work.

Minor concerns:

1. the *per2* expression seems only decreased a little bit in Supp. Fig 3C, suggesting that the dominant- negative effects of this line not sufficient.

We thank this reviewer for their comment. However, the *per2* qPCR analysis was done on whole embryos and since the dominant-negative *bmal1a* acts specifically in endothelial cells of *kdr1:GAL4;UAS:DN-Bmal1a* embryos we believe that the significant *per2* expression reduction indicates that the dominant-negative does in fact act in the manner that we expect it to, since *per2* is a direct target gene of *Bmal1a*. The following sentence has been added at the relevant place to notify the reader: “*per2* expression was not totally abolished as our dominant-negative construct was only effective in ECs, when qPCR was performed on whole embryos.”

Second decision letter

MS ID#: dev.204726R1

MS TITLE: BMAL1 modulates glutamine supply to control hematopoietic stem and progenitor cell expansion

AUTHORS: Tim Petzold, Lydia K. Lutes, Keila Navarro I. Batista, Antonia Konle, Bastien Baechler, Stéphane Jemelin, Holger Gerhardt, Rachel Golub, Christoph Scheiermann and Julien Y. Bertrand

Dear Julien,

I have now received all the referees' reports on the above manuscript, and have reached a decision. The referees' comments are appended below.

As you will see, while two of the referees are largely happy with your revisions (with the caveat that in comments to the editor, reviewer 3 said that their concerns were not fully addressed), the third reviewer considers that the critical issues remain to be addressed. The points that this reviewer raises appear to be important and so I do think that they need to be addressed. If you are able to revise the manuscript along the lines suggested, I will be happy to receive a further revised version of the manuscript.

Please attend to all of the reviewers' comments and ensure that you upload both a 'clean' version of your Word file, along with a highlighted version clearly showing where you have made changes in the revised manuscript. Please avoid using 'Tracked changes' in Word files as these are lost in PDF conversion. I should be grateful if you would also provide a point-by-point response detailing how you have dealt with the points raised by the reviewers in the 'Response to Reviewers' box. If you do not agree with any of their criticisms or suggestions please explain clearly why this is so.

Reviewer 1

Advance summary and potential significance to field

Comments for the author

All concerns from the original submission were addressed.

Reviewer 2

Advance summary and potential significance to field

Comments for the author

While this reviewer appreciates the efforts that authors have made answering to the series of issues regarding their submitted work (light/dark cycles, the potential role of *bmal1* in the regulation of

lineage-restricted progenitors (ex: erythroid progenitors such as EMPs), silencing in *gata2b* the transgenic line, ...), three essential issues have not been thoroughly and rigorously addressed.

1- Regarding the lack of expansion of differentiated cells in DN:*bmal1* larvae at 4.5 dpf (neutrophils (*mpx+*), macrophages (*mfap+*), T-cells (*rag1+*), erythrocytes (*gata1+*)), the argument developed by the authors in the revised version is quite confusing. If, as they propose, this lack of expansion is most likely explained by the fact that these differentiated cells arise from HSC-independent hematopoiesis, this infers that progenitors contained in their so-called 'HSPC' pool (hence also containing progenitor cells) contribute minimally to differentiated cells at that stage. This would mean that progenitors in this HSPC population would be HSC-dependent, the population supposedly subjected to *bmal1*-dependent regulation of proliferation in the CHT (and/or migratory capacity). This is very unlikely because it would mean that the *cmyb+* cells that expand upon DN-*bmal1a* expression (starting around 60 hpf (Fig. 2A) and in early larval stage (4.5 dpf, Fig. 2B)) would do so much earlier than HSC-dependent progenies described by others (around 8 dpf and beyond, see Ulloa et al Cell Rep 2021, Tian et al J Exp Med 2017). In addition, around 5 dpf, HSC-dependent cells are expected to represent a relatively minor fraction of populations homing in the CHT vein plexus; the hematopoietic cells homing in the CHT during the developmental period are in the vast majority erythroid progenitors (see Ulloa et al Cell Rep 2021 and Torcq et al Development 2025). Finally, this is not in agreement with the RNAseq results of this study in which none of the typical markers of the HSC-dependent pool that have been identified appear to be upregulated in DN-*Bmal1* expressing cells (ex: *draculin*, *gata2b*, see Ulloa et al 2021).

2- Regarding the RNAseq results of this paper (Fig. 4), quite importantly and if correct, the previous Table S6 that provided information on differential expression of markers between DN-*bmal1* expressing cells and control has been removed from the revised version. Why is it so?

On this line, as mentioned in my former point 3, this Table revealed that among the main markers differentially upregulated are some of the erythroid lineage (*hbbe1.1*, *hbbe3*, *hbae1* etc ...). Although the authors have made some efforts to address the question of the potential main population concerned by the *bmal1* control of proliferation which is proposed, e.g erythromyeloid progenitors (EMPs), the data are still consistent with the contribution of HSC-independent lympho-erythroid-primed progenitors (LEPs, that also express *cmyb*, a population described in Ulloa et al 2021).

Resolving what is (are) the exact population(s) sensitive to *bmal1a* regulation is an essential issue for the impact of this work, submitted to Development (and clearly not beyond the scope of this very paper).

3- Finally, the work remains very weak regarding quantitative analyses. The authors argue that quantitative results from chromogenic WISH were already published (hence necessarily validating methodologies: Cacciali et al 2021, 2022, etc ..), yet this may be acceptable for cell counting (at low density), but this is not when two other parameters are analyzed (surface area (microm²) and intensity (for the latter, because of non-linearity of the chromogenic reaction)). For surface area, this does not reflect the 3D organization of organs (the CHT, the thymus (see also the quantifications in new Fig. S8 that may not be accurate)).

Quantifications of fluorescent images are not convincing either and cannot be accurate (cells/nuclei need to be segmented, after confocal z-stack acquisitions and 3D reconstitutions), see for ex: Fig. 2C, Fig. S6 with which the authors argue that they are addressing HSPCs (HSC-dependent pool) by visualizing *CD41/eGFP+* and *runx1/nls-mCherry+* cells supposedly significantly sensitive to DN-*Bmal1a* expression in comparison to control. Lack of accuracy in quantifications is even more at high risk for the *runx1* reporter line because of low signal to noise ratio (hence requiring high resolution 3D analyses).

Reviewer 3

Advance summary and potential significance to field

Comments for the author

The authors have revised the manuscript accordingly.

Second revision

Author response to reviewers' comments

Reviewer 1: SUMMARY OF THE ADVANCE MADE IN THIS PAPER AND ITS POTENTIAL SIGNIFICANCE TO THE FIELD

SUGGESTIONS TO AUTHORS

All concerns from the original submission were addressed.

Reviewer 2: SUMMARY OF THE ADVANCE MADE IN THIS PAPER AND ITS POTENTIAL SIGNIFICANCE TO THE FIELD

SUGGESTIONS TO AUTHORS

While this reviewer appreciates the efforts that authors have made answering to the series of issues regarding their submitted work (light/dark cycles, the potential role of *bmal1* in the regulation of lineage-restricted progenitors (ex: erythroid progenitors such as EMPs), silencing in *gata2b* the transgenic line, ...), three essential issues have not been thoroughly and rigorously addressed.

1- Regarding the lack of expansion of differentiated cells in DN:*bmal1* larvae at 4.5 dpf (neutrophils (*mpx+*), macrophages (*mfap+*), T-cells (*rag1+*), erythrocytes (*gata1+*)), the argument developed by the authors in the revised version is quite confusing. If, as they propose, this lack of expansion is most likely explained by the fact that these differentiated cells arise from HSC-independent hematopoiesis, this infers that progenitors contained in their so-called 'HSPC' pool (hence also containing progenitor cells) contribute minimally to differentiated cells at that stage. This would mean that progenitors in this HSPC population would be HSC-dependent, the population supposedly subjected to *bmal1*-dependent regulation of proliferation in the CHT (and/or migratory capacity). This is very unlikely because it would mean that the *cmyb+* cells that expand upon DN-*bmal1a* expression (starting around 60 hpf (Fig. 2A) and in early larval stage (4.5 dpf, Fig. 2B)) would do so much earlier than HSC-dependent progenies described by others (around 8 dpf and beyond, see Ulloa et al Cell Rep 2021, Tian et al J Exp Med 2017). In addition, around 5 dpf, HSC-dependent cells are expected to represent a relatively minor fraction of populations homing in the CHT vein plexus; the hematopoietic cells homing in the CHT during the developmental period are in the vast majority erythroid progenitors (see Ulloa et al Cell Rep 2021 and Torcq et al Development 2025).

We thank this reviewer for their comment. Regarding our explanation for the lack of differences in differentiated cell numbers in the previous version of the manuscript, we agree that the hypothesis that we set out in the previous version may not hold true and thank the reviewer for their explanation as to why this may be the case. We have therefore decided to remove this suggestion from our manuscript to avoid confusion for the reader.

We believe however, that there are a number of other potential explanations for the lack of difference observed regarding differentiated cells. For example, it may be that the expanded HSPCs differentiate later (differentiated cell numbers were quantified at 4.5 dpf) or it could also be that they do not have the same differentiation potential. One further hypothesis is that the HSPC expansion rate exceeds the differentiation rate. While these remain hypotheses which would be interesting to test in the future, we have amended the text of the manuscript with these suggestions in place of the previous one. We thank the reviewer again for pointing out the potential pitfall with our previous suggestion in the text. Together, we believe this has improved our manuscript.

Finally, this is not in agreement with the RNAseq results of this study in which none of the typical markers of the HSC-dependent pool that have been identified appear to be upregulated in DN-*Bmal1* expressing cells (ex: *draculin*, *gata2b*, see Ulloa et al 2021).

The RNA-sequencing in this manuscript was performed on FACS sorted endothelial cells and not hematopoietic cells. This explains why none of the typical markers of the HSC-dependent pool are changed in expression in our RNA-seq dataset. We do however also understand that their comment here was made in relation to the point above regarding differentiated cells and hope that the reviewer appreciates the changes / additions that we have now made to the manuscript regarding this, as outlined above.

2- Regarding the RNAseq results of this paper (Fig. 4), quite importantly and if correct, the previous Table S6 that provided information on differential expression of markers between DN-bmal1 expressing cells and control has been removed from the revised version. Why is it so?

This is indeed correct - we apologize for the confusion caused. The RNA-seq data in the table in the first version was indeed correct. For the second version, we had decided that we will upload this RNA-seq data to the gene expression omnibus so that the table of differentially expressed genes can be downloaded separately by readers. We have now reverted this and have added the RNA-seq data back to the supplemental information of the manuscript (Supplementary Table 5). We apologize for the confusion that this caused.

On this line, as mentioned in my former point 3, this Table revealed that among the main markers differentially upregulated are some of the erythroid lineage (hbbe1.1, hbbe3, hbae1 etc ...). Although the authors have made some efforts to address the question of the potential main population concerned by the bmal1 control of proliferation which is proposed, e.g erythromyeloid progenitors (EMPs), the data are still consistent with the contribution of HSC-independent lympho-erythroid-primed progenitors (LEPs, that also express cmyb, a population described in Ulloa et al 2021).

Resolving what is (are) the exact population(s) sensitive to bmal1a regulation is an essential issue for the impact of this work, submitted to Development (and clearly not beyond the scope of this very paper).

Importantly, and as also mentioned above, our RNA-sequencing was performed on endothelial cells, not hematopoietic cells. Therefore, the increase in hemoglobin gene expression in our RNA-seq does not, in our opinion, reflect any change in the composition of hematopoietic cell types present in the niche. Rather, we think that these transcriptional changes likely reflect the fact that endothelial cells can also express hemoglobin and hemoglobin related genes (Straub et al, Nature 2012, Saha et al, Int J Infl. 2014, Sangwung et al, Vasc. Med, 2017, Alvarez et al, Am. J. Respir. Cell Mol. Biol, 2017, Lechauve et al, J. Clin. Investig, 2018, Stevenson Keller, Nature Communications, 2022 and reviewed in Prabhodh S. Abbineni et al, Blood, 2024), which, as was correctly pointed out, are more highly expressed in ECs from endothelial-specific dominant-negative BMAL1A embryos. Usually, hemoglobin genes are not highly expressed in endothelial cells. Our data point towards a role of endothelial cell BMAL1A in repressing the expression of hemoglobin genes in ECs, at least at particular developmental timepoints, such as at 36 hpf, the stage at which the RNA-seq was performed.

In the past few years, many reports have demonstrated the heterogeneity of HSPC pool, mainly based on the analysis a posteriori of single cell RNA-sequencing performed on the progeny of lineage-traced HSPCs. Therefore, at this point, a variety of markers to allow discrimination of these different HSPCs is unfortunately still lacking, which would allow us to allocate the observed expansion phenotype to a particular progenitor subtype(s).

The data in our manuscript robustly demonstrates, using 4 HSPC transgenic reporter lines (including new data using the *gata2b:kalta4;UAS:GFP* line), in combination with WISH experiments, that the definitive HSPC pool size is affected in these animals. We agree that our manuscript would benefit from the language surrounding the cells we have investigated being more precise so that the reader can better appreciate which cells are affected by the pathway we have identified. Hence, to clarify the terminology, we have amended our introduction, to explain the different waves of hematopoiesis - the existence of primitive waves (both myeloid and erythroid) and the EMP wave as well as the definitive wave. Cells from the primitive and EMP waves are not derived from the aorta and are Notch-independent. Then there are at least two waves generated from the aortic floor, in a

notch-dependent way, these are the lymphoid-primed progenitors (Tian. JEM 2017) and “multipotent” progenitors, all of which are called HSPCs, and a minority of these might be true HSCs (Henninger et al, Nat Cell Biol 2016). Among these HSPCs, lineage-tracing has then shown the existence of more restricted progenitors (Ulloa et al, 2021, Ghersi et al, 2023, Xia et al, 2023, Torcq et al, 2025).

In our previous response to the reviewers, we included results performed with notch-deficient mindbomb (*mib*) mutants which are known to possess normal primitive hematopoiesis as well as normal EMP numbers. We had investigated *mpx*⁺ cell numbers in the *mib* mutants and we included the data in the previous rebuttal. Now, we have performed an additional round of experiments with the *mib* mutants, the results of which are shown in supplementary figure 11A and B. We show that while the addition of glutamine increases *cmyb* expression in the niche of *mib* siblings relative to controls (Supplementary Figure 11A), no effect on *cmyb* expression levels and thus HSPC numbers was present in the niche of mindbomb mutants (Supplementary Figure 11B). This demonstrates that the phenotype concerns aortic-derived notch-dependent HSPCs. Please note, in these experiments the phenotypes were so evident from the WISH images that we decided to quantify and present the number of embryos which looked the same as the representative image in each case, which are the fractions shown on the images (and explained in the figure legend).

Additionally, we have now also performed additional experiments using the *gata2b:kalta4;UAS:GFP* line, which we had previously not used in this study. In this line, only progenitors from the definitive wave are marked. We were unable to cross *gata2b:kalta4;UAS:GFP* with *kdr1:GAL4;UAS:DN-bmal1a* fish, since in the offspring embryos GAL4 would activate UAS:GFP in all endothelial cells making it impossible for us to quantify the number of *gata2b:GFP*⁺ cells in the CHT region. Therefore, instead we treated *gata2b:kalta4;UAS:GFP* embryos with 1mM Glutamine or 1mM L-leucine (a GLUD1A activator) from 48 to 72 hpf. At 72 hpf, we quantified the number of *gata2b:GFP*⁺ cells in the CHT. We found that the addition of glutamine resulted in a significant increase in the number of *gata2b:GFP*⁺ cells in the CHT (Supplementary Figure 11C), while the addition of L-leucine resulted in a decrease in the number of *gata2b:GFP*⁺ cells in the CHT (Supplementary Figure 11C). These results corroborate data throughout our manuscript and highlight that expansion of definitive HSPCs is affected in the presence of EC-specific DN-*bmal1a*.

In the future, we agree that following the identification of markers of all of the HSPC subtypes it will be very interesting to subsequently determine precisely which HSPC subsets (if not all of them) are affected in EC specific DN-*bmal1a* embryos. We have changed the text of this manuscript to specify that the *bmal1a-glud1*-glutamine pathway affects definitive, aortic-derived progenitors. We do however understand and agree with the reviewers point regarding the interest in better understanding which HSPC subtypes are affected and so we have now also acknowledged this in the discussion of our manuscript.

3- Finally, the work remains very weak regarding quantitative analyses. The authors argue that quantitative results from chromogenic WISH were already published (hence necessarily validating methodologies: Cacciali et al 2021, 2022, etc ..), yet this may be acceptable for cell counting (at low density), but this is not when two other parameters are analyzed (surface area (microm²) and intensity (for the latter, because of non-linearity of the chromogenic reaction)). For surface area, this does not reflect the 3D organization of organs (the CHT, the thymus (see also the quantifications in new Fig. S8 that may not be accurate)).

While we agree that the analysis referred to by this reviewer would be enhanced by 3D imaging of the CHT, thymus and kidney glomeruli, since, as the reviewer mentions, they are of course 3D structures. However, we are convinced that the methods and approaches that we have used to undertake the quantifications referred to remain valid to generate a robust readout of the average number of cells / average signal area (which in turn provides information on the number of relevant cells present) per embryo.

To answer the reviewer’s specific points, they mention 2 parameters, signal intensity and surface area, which they state may make WISH quantifications such as those we have performed inaccurate. Regarding surface area, we do agree that the analysis referred to by this reviewer would be enhanced by 3D imaging of the CHT, thymus and kidney glomeruli, as mentioned. However, we did not do this, since we believe that gene expression 2D quantification is a reliable

proxy, therefore robust and informative. 2D *in situ* hybridization signals has also been quantified for 3D structures to measure gene expression differences between conditions in this way in many prior publications by labs undertaking research in our field including the North lab (Frame et al, Dev Cell, 2020, Lundin et al, Dev Cell, 2020, Soto et al, Stem Cell Reports, 2021), the Bowman and Targoff labs (Development, 2026), the Robin lab (Klaus et al, Cell Reports 2022), the Nicoli lab (Gherzi et al, Nature Cell Biology 2023, Kasper et al, Science, 2020), as well as previous reports from our lab (Cacialli et al, Nature Communications, 2021). We have now added references to some of these publications in the relevant section of the materials and methods. Together, we believe it was a valid approach to use in our study and provides results that we believe are reliable and robust.

Regarding the second point, signal intensity differences which may arise due to the non-linearity of the chromogenic reaction, we would like to re-iterate that while staining always differs between *in situ* hybridization experiments performed, the staining period is always kept the same between different conditions in one particular experiment, allowing comparisons to be made between the conditions. Hence, while it is true that signal intensity may not always correlate perfectly with expression of a gene, due to the lack of linear development of chromogenic signal (as stated by the reviewer), we believe it is still valid to use this approach to compare expression levels between different conditions that have been treated side by side.

However, since this reviewer pointed this out as a potential issue, we decided we would like investigate this further. Hence, we decided to re-analyse some of the WISH signal intensity for data already present in our manuscript. We quantified the WISH signal intensity for the data used to generate Figure 2B, in which we showed that the *cmym* signal area is significantly larger at 4.5 dpf when DN-bmal1a is expressed in the endothelium. We have now quantified the *cmym* signal intensity (pixel intensity) of these embryos (images) in exactly the same region and found that there is no difference in the signal intensity (see data below), indicating that the signal intensity is not different across batches of embryos between conditions. We hope that this satisfies this reviewer regarding this point.

Quantifications of fluorescent images are not convincing either and cannot be accurate (cells/nuclei need to be segmented, after confocal z-stack acquisitions and 3D reconstitutions), see for ex: Fig. 2C, Fig. S6 with which the authors argue that they are addressing HSPCs (HSC-dependent pool) by visualizing CD41/eGFP+ and runx1/nls-mCherry+ cells supposedly significantly sensitive to DN-Bamal1a expression in comparison to control. Lack of accuracy in quantifications is even more at high risk for the runx1 reporter line because of low signal to noise ratio (hence requiring high resolution 3D analyses).

We agree with this reviewer that high resolution 3D analyses would have enhanced our work. As such we have now highlighted this as a new point in the discussion of our manuscript to acknowledge this.

However, we also believe that use of the 4 different transgenic HSPC-marker lines used throughout this study provide robust and convincing evidence for the phenotype that is present, along with all of the WISH data which corroborate our findings using the transgenic lines.

We have decided to invert the representative CHT images for the experiments in which we performed the quantifications of HSPC numbers in the CHT. We have also made them black and white. We believe this allows the reader to better appreciate the differences present between genotypes, which, we agree was at times somewhat difficult to do with the color images previously. We also appreciate the challenges faced in the field when using fluorescent reporter lines marking HSPCs and hence, this is the reason we decided to corroborate our findings using the *cmyb*:GFP line with data from 3 further lines, the *cd41*:EGFP line, the *runx1*:nls-mCherry line, and, as mentioned above we have now included data using the *gata2b*:GFP line as well (Supplementary Figure 11C). While we agree that the *runx1* reporter line used can have a low signal to noise ratio, we found that we were able to accurately count HSPCs in images of the *kdrl*:nls-EGFP, *cmyb*:GFP, *cd41*:EGFP, *runx1*:nls-mCherry and *gata2b*:GFP lines by using the counter tool following image inversion, sharpening and enlarging the region of interest of each image in Fiji ImageJ, as has been done in our lab previously (Cacialli et al, Nature Communications, 2021; Cacialli et al, EMBO Journal, 2022) as well as in work from many other labs (Liu et al, Development 2025; Schiavo and Tamplin, Development, 2022; Soto et al, Stem Cell Reports, 2021; Frame et al, Developmental Cell, 2020; Lundin et al, Developmental Cell, 2020; Brix et al, Communications Biology 2024; Bornhorst et al, Nature Communications, 2024). We have added some of these references to our manuscript in the relevant section of the materials and methods.

We hope that these explanations and alterations to the text of the manuscript satisfy the reviewer and we thank them again as we believe our manuscript has been improved through their suggestions.

Reviewer 3: SUMMARY OF THE ADVANCE MADE IN THIS PAPER AND ITS POTENTIAL SIGNIFICANCE TO THE FIELD

SUGGESTIONS TO AUTHORS

The authors have revised the manuscript accordingly.

Third decision letter

MS ID#: dev.204726R2

MS TITLE: BMAL1 modulates glutamine supply to control hematopoietic stem and progenitor cell expansion

AUTHORS: Tim Petzold, Lydia K. Lutes, Keila Navarro I. Batista, Antonia Konle, Bastien Baechler, Stéphane Jemelin, Holger Gerhardt, Rachel Golub, Christoph Scheiermann and Julien Y. Bertrand

Dear Julien,

I sent your manuscript back to reviewer 2 who is now happy for the study to be published. Consequently, your manuscript has been accepted for publication in Development, pending our standard publication integrity checks.

Reviewer 2

Advance summary and potential significance to field

Comments for the author

This reviewer thanks the authors for the time spent on additional revision of their work and manuscript. The results added with the *gata2b* fish line (new Supplemental Fig.11) reinforce their conclusions on the proposed function of *bmal1a* in HSPC expansion. Overall, with clarified text, the manuscript has improved significantly.